# Differentially Private Optimizers Can Learn Adversarially Robust Models

**Zhiqi Bu**[*]                                                                *woodyx218@gmail.com*

**Yuan Zhang**[*]                                                             *ewyuanzhang@gmail.com*

**Reviewed on OpenReview:** *https://openreview.net/forum?id=o8VgRNYh6n*

## Abstract

Machine learning models have shone in a variety of domains and attracted increasing attention from both the security and the privacy communities. One important yet worrying question is: Will training models under the differential privacy (DP) constraint have an unfavorable impact on their adversarial robustness? While previous works have postulated that privacy comes at the cost of worse robustness, we give the first theoretical analysis to show that DP models can indeed be robust and accurate, even sometimes more robust than their naturally-trained non-private counterparts. We observe three key factors that influence the privacy-robustness-accuracy tradeoff: (1) hyper-parameters for DP optimizers are critical; (2) pre-training on public data significantly mitigates the accuracy and robustness drop; (3) choice of DP optimizers makes a difference. With these factors set properly, we achieve 90% natural accuracy, 72% robust accuracy (+9% than the non-private model) under $l_2(0.5)$ attack, and 69% robust accuracy (+16% than the non-private model) with pre-trained SimCLRv2 model under $l_\infty(4/255)$ attack on CIFAR10 with $\epsilon = 2$. In fact, we show both theoretically and empirically that DP models are Pareto optimal on the accuracy-robustness tradeoff. Empirically, the robustness of DP models is consistently observed across various datasets and models. We believe our encouraging results are a significant step towards training models that are private as well as robust.

## 1 Introduction

Machine learning models trained on large amount of data can be vulnerable to privacy attacks and leak sensitive information. For example, Carlini et al. (2021) shows that attackers can extract text input from the training set via GPT2 (Radford et al., 2019), that contains private information such as address, phone number, name and so on; Zhu et al. (2019) shows that attackers can recover both the image input and the label from gradients of ResNet (He et al., 2016) trained on CIFAR100 (Krizhevsky et al., 2009), SVHN (Netzer et al., 2011), and LFW (Huang et al., 2008).

To protect against the privacy risk rigorously, the differential privacy (DP) is widely applied in various deep learning tasks (Abadi et al., 2016; McMahan et al., 2017; Bu et al., 2020; Nori et al., 2021; Li et al., 2021), including but not limited to computer vision, natural language processing, recommendation system, federated learning and so on. At high level, the privacy is protected via DP optimizers such as DP-SGD and DP-Adam, while allowing the models to remain highly accurate. In other words, the privacy concerns have been largely alleviated by switching from regular optimizers to DP ones.

---

[*]Equal contribution.

An equally important concern from the security community is that, many models such as deep neural networks are known to be vulnerable against adversarial attacks. This robustness risk can be severe when the attackers can successfully fool models to make the wrong prediction, through modifying the input data by a negligible amount. An example from Engstrom et al. (2019) shows that a strong ResNet50 trained on ImageNet (Deng et al., 2009) with 76.13% accuracy can degrade to 0.00% accuracy, even if the input image is merely perturbed by 4/255 at each pixel.

However, at the intersection of these two concerns, previous works have empirically observed an upsetting privacy-robustness tradeoff under some scenarios, implying the implausibility of achieving both robustness and privacy at the same time. In Song et al. (2019) and Mejia et al. (2019), adversarially trained models are shown to be more vulnerable to privacy attacks, such as the membership inference attack, than naturally trained models. In Tursynbek et al. (2020) and Boenisch et al. (2021), DP trained models were more vulnerable to robustness attacks than naturally trained models on MNIST and CIFAR10. This leads to the following concern:

Does DP optimization necessarily lead to less adversarially robust models?

On the contrary, we show that DP models can be adversarially robust, sometimes even more robust than the naturally trained models. Indeed, we illustrate that DP models can be Pareto optimal, so that any model with higher accuracy than DP ones must have worse robustness. We observe that:

1. DP training itself does not worsen adversarial robustness in comparison to the natural training.

2. The robustness is largely affected by the DP optimization hyper-parameters $(R, \eta)$, where different hyper-parameters are equally privacy-preserving but significantly different in accuracy and robustness.

3. DP optimization hyper-parameters $(R, \eta)$ achieving the best accuracy (which can be much less robust than the natural training) is different to those achieving the best robustness.

4. The robustness is less affected by the privacy budget (e.g. Table 4 and Table 6). Even when we consider non-DP optimization (i.e. $\epsilon = \infty$, no noise but with clipping), the models could be comparably or more robust than naturally trained models (see the blue boxplots in Figure 3).

In sharp contrast to the empirical nature of previous arts, we enhance our understanding about the adversarial robustness of DP models from a theoretical angle. Our analysis shows that DP classifiers without adversarial training may in fact be the most adversarially robust classifier. Motivated by our theoretical analysis, we claim that the hyperparameter tuning is vital to successfully learning a robust and private model, where the optimal choice is to use small clipping norms and large learning rates. This is interesting as such a hyperparameter choice is also observed to be the most effective in learning highly accurate models under DP (Li et al., 2021; Kurakin et al., 2022; De et al., 2022). In fact, using a small clipping norm is equivalent to normalizing all per-sample gradients, which allows a clear demonstration in Table 1 that the optimal hyperparameter for natural accuracy is different to that for robust accuracy (only $\eta$ is present because $R$ is absorbed in the automatic DP-SGD (Bu et al., 2023b)).

Table 1: Robust and natural accuracy are achieved by different hyperparameter $\eta$ on CIFAR10. Same setting as in Tramer & Boneh (2020), under $(\epsilon, \delta) =$(2,1e-5) and attacked by 20 steps of $l_\infty(2/255)$ PGD.

| | learning rate $\eta$ | $2^{-8}$ | $2^{-7}$ | $2^{-6}$ | $2^{-5}$ | $2^{-4}$ | $2^{-3}$ | $2^{-2}$ | $2^{-1}$ |
|---|---|---|---|---|---|---|---|---|---|
| DP | natural accuracy | 84.36 | 87.33 | 89.45 | 90.76 | 91.76 | 92.50 | 92.54 | **92.70** |
| | robust accuracy | 75.45 | 78.92 | **81.03** | 80.96 | 78.87 | 72.77 | 58.29 | 26.97 |
| | learning rate $\eta$ | $2^{-5}$ | $2^{-4}$ | $2^{-3}$ | $2^{-2}$ | $2^{-1}$ | $2^{0}$ | $2^{1}$ | $2^{2}$ |
| non-DP | natural accuracy | 94.23 | 94.31 | 94.38 | 94.46 | **94.60** | 94.39 | 94.21 | 93.84 |
| | robust accuracy | 79.32 | **79.57** | 74.18 | 66.20 | 55.87 | 46.00 | 41.75 | 45.21 |

Additionally, we advocate pretraining the model and selecting proper optimizers for DP training, which allow us to max out the performance on MNIST (LeCun et al., 1998), Fashion MNIST(Xiao et al., 2017), CIFAR10(Krizhevsky et al., 2009), and CelebA(Liu et al., 2015).

*Remark* 1. Most of this work does not use adversarial training (except in Table 2 and Table 3) and should be distinguished from the certified robustness (Lecuyer et al., 2019), which does not guarantee DP in a per-sample sense rather than use the mathematical tools from DP in a per-pixel way.

## 2 Preliminaries

**Notation.** We use $f : \mathcal{X} \to \mathcal{Y}$ to denote the model mapping from data space $\mathcal{X}$ to label space $\mathcal{Y}$. We denote the datapoints as $\{\boldsymbol{x}_i\} \in \mathbb{R}^d$ and the labels as $\{y_i\}$, following i.i.d. from the distributions $\boldsymbol{x}$ and $y$ respectively. We denote the gradient of the $i$-th sample at step $t$ as $g_t(\boldsymbol{x}_i, y_i; \mathbf{w}, b)$, where $\mathbf{w}$ is the weights and $b$ is the bias of model $f$.

To start, we introduce the definition of DP, particularly the $(\epsilon, \delta)$-DP (Dwork et al., 2014).

**Definition 1.** A randomized algorithm $M$ is $(\varepsilon, \delta)$-DP if for any neighboring datasets $S, S'$ that differ by one arbitrary sample, and for any event $E$, it holds that

$$\mathbb{P}[M(S) \in E] \leqslant e^\varepsilon \mathbb{P}\left[M\left(S'\right) \in E\right] + \delta. \tag{1}$$

In words, DP guarantees in the worst case that adding or removing one single datapoint $(\boldsymbol{x}_i, y_i)$ does not affect the model much, as quantified by the small constants $(\epsilon, \delta)$. Therefore DP limits the information possibly leaked from such datapoint.

In deep learning, DP is guaranteed by privatizing the gradient in two steps: (1) the per-sample gradient clipping[1] (specified by the clipping norm $R$, to bound the sensitivity of $\sum \boldsymbol{g}_t(x_i)$); (2) the random noising (specified by the noise multiplier $\sigma_{\text{DP}}$, to randomize the outcome so that each sample's contribution is indistinguishable; $\sigma_{\text{DP}}$ can be determined by the privacy accountants Dwork et al. (2014); Abadi et al. (2016)). From an algorithmic viewpoint, DP training simply applies any optimizer on the private gradients instead of on the regular gradients.

$$\text{Non-DP training on regular gradient:} \qquad\qquad\qquad \sum_i \boldsymbol{g}_t(\boldsymbol{x}_i, y_i) \tag{2}$$

$$\text{DP training on private gradient:} \qquad \sum_i C_R(\boldsymbol{g}_t(\boldsymbol{x}_i, y_i)) + \sigma_{\text{DP}} R \cdot \mathcal{N}(\mathbf{0}, \boldsymbol{I}) \tag{3}$$

We are interested in the adversarial robustness of models trained by DP optimizers. To be sure, we consider the adversarially robust classification error and the natural classification error as

$$\mathcal{R}_\gamma(f) := \mathbb{P}(\exists ||\boldsymbol{p}||_\infty < \gamma, \text{ s.t. } f(\boldsymbol{x} + \boldsymbol{p}) \neq y), \quad \mathcal{R}_0(f) := \mathbb{P}(f(\boldsymbol{x}) \neq y). \tag{4}$$

where $\boldsymbol{p} \in \mathbb{R}^d$ is the adversarial perturbation, $\gamma$ is the attack magnitude, and $l_\infty$ (and $l_2$) attack is considered. Notice that when $\gamma = 0$, the robust error in (4) reduces to the natural error.

## 3 Theoretical Analysis on Linear Classifiers

To theoretically understand the adversarial robustness of DP learning, we study the robustness of the DP and non-DP linear models on a binary classification problem. We consider a mixed Gaussian distribution, where the positive class $y = +1$ has a larger variance (i.e. it is more difficult to be classified correctly[2]) than the negative class $y = -1$:

$$\boldsymbol{x} \sim \begin{cases} \mathcal{N}\left(\boldsymbol{\theta}_d, K^2\sigma^2 \mathbf{I}_d\right) & \text{if } y = +1 \\ \mathcal{N}\left(-\boldsymbol{\theta}_d, \sigma^2 \mathbf{I}_d\right) & \text{if } y = -1 \end{cases} \tag{5}$$

---

[1]We use $C_R(g_t(x_i)) := g_t \cdot \min\{R/||g_t||_2, 1\}$ as in Abadi et al. (2016) to denote the gradient clipping, after which each per-sample gradient has norm $\leq R$. Note that in Table 1 we use the automatic clipping such that $C_R(g_t(x_i)) := g_t/||g_t||_2$.

[2]The fact that larger variance indicates lower intra-class accuracy is proven by Xu et al. (2021, Theorem 1).

where $y \overset{\text{unif}}{\sim} \{-1, +1\}$, $\boldsymbol{\theta}_d = (\theta, \cdots, \theta) \in \mathbb{R}^d$, $\sigma > 0$ and $K > 1$.

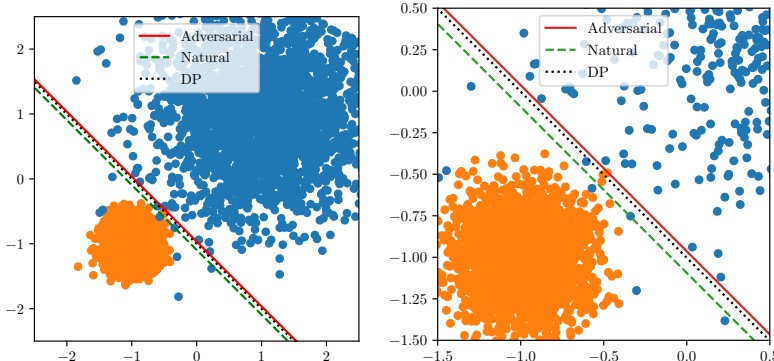

Figure 1: Decision boundaries of linear classifiers for (5), $K = 4, \sigma = 0.2, \theta = 1$.

The setting[3] in (5) is analyzable because of the data symmetry along the diagonal axis $\mathbb{E}x_1 = \cdots = \mathbb{E}x_d$, as illustrated in Figure 1. We will show that this symmetry leads to an explicit decision hyperplane of linear classifiers in Theorem 1, which further characterizes the strongest adversarial perturbation $\boldsymbol{p}^* \equiv \arg\sup_{\|\boldsymbol{p}\|_\infty < \gamma} \mathbb{P}(f(\boldsymbol{x} + \boldsymbol{p}) \neq y)$ explicitly.

In the following analysis, we focus on the linear classifiers $f(\boldsymbol{x}; \mathbf{w}, b) = \text{sign}(\sum_{j=1}^d w_j x_j + b)$ with weights $w_j$ and bias $b$.

*Remark* 2. Within the family of linear classifiers, by the symmetry of data in (5), it can be rigorously shown by (6) that the optimal weights with respect to the natural and robust errors are always $w_1 = \cdots = w_d$, which matches our empirical calculation on DP models. That is, the weights do not distinguish between the robust and natural models. Consequently, the key to the adversarial robustness lies in the intercept $b$, which is analyzed in the subsequent sections.

## 3.1 Optimal Robust and Natural Linear Classifiers

We start by reviewing the robust error of robust classifier and the explicit formula of its intercept.

**Theorem 1** (Extended from Theorem 2 in Xu et al. (2021)). *For data distribution $(\boldsymbol{x}, y)$ in Equation (5) and under the $\gamma$ attack magnitude, we define the optimal robust linear classifier as*

$$f_\gamma = \underset{f \text{ is linear}}{\arg\min} \ \mathbb{P}(\exists \|\boldsymbol{p}\|_\infty < \gamma, \ s.t. \ f(\boldsymbol{x} + \boldsymbol{p}) \neq y) = \underset{f \text{ is linear}}{\arg\min} \ \mathcal{R}_\gamma(f).$$

*The optimal robust error is*

$$\mathcal{R}_\gamma(f_\gamma) = \frac{1}{2}\Phi\left(B(K, \gamma) - K\sqrt{B(K, \gamma)^2 + q(K)}\right) + \frac{1}{2}\Phi\left(-KB(K, \gamma) + \sqrt{B(K, \gamma)^2 + q(K)}\right),$$

*where $\Phi$ is the cumulative distribution function of standard normal, $B(K, \gamma) = \frac{2}{K^2-1}\frac{\sqrt{d}(\theta - \gamma)}{\sigma}$ and $q(K) = \frac{2\log K}{K^2-1}$. Furthermore, by the symmetry of the data distribution, we have*

$$1, \cdots, 1, b_\gamma = \underset{\mathbf{w}, b}{\arg\min} \mathcal{R}_\gamma(f(\cdot; \mathbf{w}, b)), \tag{6}$$

$$b_\gamma = \frac{K^2+1}{K^2-1}d(\theta - \gamma) - K\sqrt{\frac{4d^2(\theta - \gamma)^2}{(K^2-1)^2} + d\sigma^2 q(K)}. \tag{7}$$

---

[3]This setting is also studied in Xu et al. (2021), which focuses on the robustness-fairness tradeoff, and thus is different to our interest.

Theorem 1 gives the closed form of the optimal robust classifier $f_\gamma$, or equivalently its intercept $b_\gamma$, and the optimal robust error. The special case of natural error can be easily recovered by setting $\gamma = 0$:

$$1, \cdots, 1, b_0 = \arg\min_{\mathbf{w}, b} \mathcal{R}_0(f(\cdot; \mathbf{w}, b)).$$

We know for sure from Theorem 1 that there exists a tradeoff between robustness and accuracy: it is impossible for the natural classifier $f_0$ to be optimally robust or the robust classifier $f_\gamma$ to be optimally accurate, since $b_0 \neq b_\gamma$ (c.f. Figure 2).

**Fact 1.** $b_\gamma$ in (7) is strictly decreasing in $\gamma$, ranging from $b_0$ to $-\infty$.

*Proof of Fact 1.* Proof 5 in Xu et al. (2021) shows that $\frac{db_\gamma}{d\gamma} \leq -\frac{K-1}{K+1}d < 0$, thus $b_\gamma$ is strictly decreasing in $\gamma$. Therefore, the range of $b_\gamma$ is $(b_\infty, b_0]$. Finally, we note that $b_\gamma < \frac{K^2+1}{K^2-1}(\theta - \gamma)$, hence $b_\infty = -\infty$. □

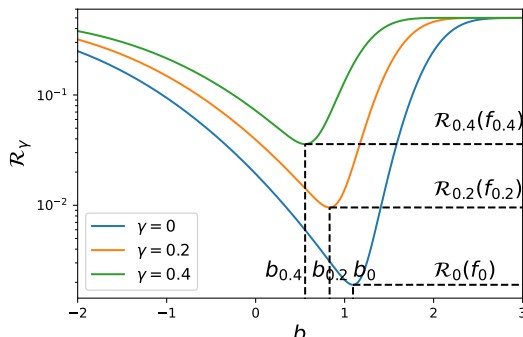

Figure 2: Intercepts and robust/natural accuracy under (5), with $K = 4, \sigma = 0.2, \theta = 1$.

## 3.2   Adversarially Robust Errors of Private Linear classifiers

Now we analyze the robust error of DP classifiers, which requires a different analysis from Theorem 1 because $\langle 1 \rangle$ DP modifies the optimization instead of the objective function, unlike $\mathcal{R}_\gamma$ which modifies the natural error $\mathcal{R}_0$; $\langle 2 \rangle$ consequently, deriving the DP parameters $\mathbf{w}$ and $b$ is much more difficult: we need to solve $\arg\min_{\mathbf{w}, b} \mathcal{R}_\gamma(f(\cdot; \mathbf{w}, b))$ under the additional constraint (imposed by DP) that (3) equals 0 in expectation. While it is possible to simplify the problem, for example, by analyzing the deterministic gradient flow Bu et al. (2023a) or by formulating the actual objective that DP-SGD optimizes Song et al. (2021), this is beyond the scope of this work.

We consider a specific linear classifier $f(\cdot; \mathbf{1}, b)$ where only the intercept $b$ is learned and privatized[4]. This is known as the bias term fine-tuning (Zaken et al., 2022; Bu et al., 2022b), a popular approach in fine-tuning the DP or standard neural networks.

For this classifier and any $b$, the robust error ($\gamma \neq 0$) and the natural error ($\gamma = 0$) are:

$$\mathcal{R}_\gamma(f) = \mathbb{P}(\exists \|\boldsymbol{p}\|_\infty \leq \gamma \text{ s.t. } f(\boldsymbol{x} + \boldsymbol{p}) \neq y) = \max_{\|\boldsymbol{p}\|_\infty \leq \gamma} \mathbb{P}(f(\boldsymbol{x} + \boldsymbol{p}) \neq y)$$

$$= \frac{1}{2}\mathbb{P}(f(\boldsymbol{x} + \boldsymbol{\gamma}_d) \neq -1 \mid y = -1) + \frac{1}{2}\mathbb{P}(f(\boldsymbol{x} - \boldsymbol{\gamma}_d) \neq +1 \mid y = +1)$$

$$= \frac{1}{2}\mathbb{P}\left(\sum_{i=1}^d w_j (x_j + \gamma) + b > 0 \mid y = -1\right) + \frac{1}{2}\mathbb{P}\left(\sum_{i=1}^d w_j (x_j - \gamma) + b < 0 \mid y = +1\right)$$

$$= \frac{1}{2}\Phi\left(-\frac{\sqrt{d}(\theta - \gamma)}{\sigma} + \frac{1}{\sqrt{d}\sigma} \cdot b\right) + \frac{1}{2}\Phi\left(-\frac{\sqrt{d}(\theta - \gamma)}{K\sigma} - \frac{1}{K\sqrt{d}\sigma} \cdot b\right) \qquad (8)$$

where $\boldsymbol{\gamma}_d \equiv (\gamma, \cdots, \gamma)$. With (8), we can analyze the robust and natural errors for any intercept $b$ (private or not, robust or natural) and any attack magnitude $\gamma$.

Our next result answers the following question: fixing a DP classifier $f_{\text{DP}} := f(\cdot; \mathbf{1}, b_{\text{DP}})$, or equivalently its intercept $b_{\text{DP}}$, under which attack magnitude is the classifier robust? We show that, although $b_{\text{DP}}$ is not available in the closed form, it is possible for some attack magnitude $\gamma^*$ that $b_{\text{DP}} = b_{\gamma^*}$, and thus the DP classifier is the most robust classifier among all.

---

[4]Note that DP is only required on trainable parameters that are learned from data; otherwise no data privacy can be leaked. Therefore this specific classifier is guaranteed to be DP.

**Theorem 2.** *For data distribution $(\boldsymbol{x}, y)$ in Equation (5) and for any $b_{\mathrm{DP}} < b_0$, there exists $\gamma^* > 0$ such that $b_{\gamma^*} = b_{\mathrm{DP}}$, and therefore*

$$\min_{f \ is \ linear} \mathcal{R}_{\gamma^*}(f) \equiv \mathcal{R}_{\gamma^*}(f_{\gamma^*}) = \mathcal{R}_{\gamma^*}(f_{\mathrm{DP}}).$$

*Proof of Theorem 2.* By Fact 1, $b_\gamma - b_{\mathrm{DP}}$ is decreasing in $\gamma$, ranging from $b_0 - b_{\mathrm{DP}}$ to $-\infty$. By the intermediate value theorem, there exists $\gamma^* > 0$ such that $b_{\gamma^*} = b_{\mathrm{DP}}$, i.e. $f_{\mathrm{DP}} = f_{\gamma^*}(\cdot; \mathbf{1}, b_{\gamma^*})$. $\qquad\square$

By Theorem 2, as long as the DP intercept is sufficiently small, the DP classifier must be the most robust under some attack magnitude, among all linear classifiers. We visualize in Figure 3 at $\gamma^* = 0.075$, that indeed $b_{\mathrm{DP}} \approx b_{\gamma^*}$ (grey solid line).

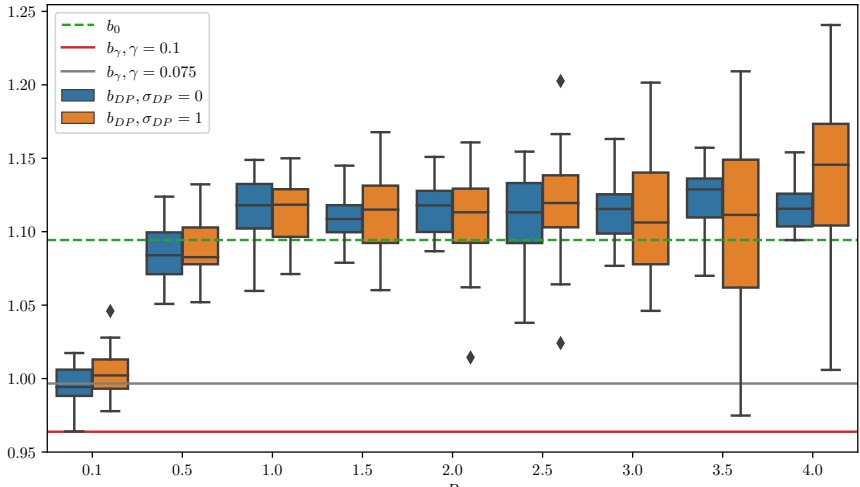

Figure 3: Intercepts (decision boundaries) for (5), same setup as Figure 2. For DP classifiers, we use DP-SGD with $\eta = 8$, epochs=50, batch size=1000, sample size=10000, $(\epsilon, \delta) = (15, 1e-4)$.

To validate the condition of Theorem 2, we now demonstrate the achievability of $b_{\mathrm{DP}} < b_0$ as a result of the per-sample gradient clipping in DP optimizers. In words, we show that the robust intercept is stationary by the DP gradient descent but not so by the regular gradient descent.

We consider the above setting with an attack magnitude $\gamma$ and $\sigma_{\mathrm{DP}} = 0$. We temporarily ignore the noise because it only adds variance to the gradient but does not affect the mean (see the blue and orange boxes in Figure 3), and when learning rate is small, $\sigma_{\mathrm{DP}}$ has little effect on convergence (Bu et al., 2023a).

In DP gradient descent, if the clipping norm $R$ is sufficiently small, then each per-sample gradient $\boldsymbol{g}_t(\boldsymbol{x}_i, y_i; \mathbf{1}, b_\gamma)$ has the same magnitude $R$ after clipping. Therefore the positive samples, pushing the intercept $b$ to increase, can balance with the negative ones that pull $b$ to descrease. Thus $b = b_\gamma$ is a stationary point learnable by DP training, even though it is smaller than $b_0$ by Fact 1. However, in the non-DP gradient descent, $b_\gamma$ is not stationary. This is because the positive class gradient $\sum_i g_t(\boldsymbol{x}_i, +1; \mathbf{1}, b_\gamma)$ is larger than the negative class gradient $\sum_i g_t(\boldsymbol{x}_i, -1; \mathbf{1}, b_\gamma)$, so as to push the decision boundary $b_\gamma$ towards $b_0$, where the natural classifier $f_0$ is defined. We visualize our analysis in Figure 4.

Next, suppose we only require the DP classifier to be more robust than the natural classifier, without requiring it to be the most robust among all linear classifiers. We can answer the question: fixing the attack magnitude $\gamma$, under which condition is $f_{\mathrm{DP}}$ more robust than the natural classifier $f_0$?

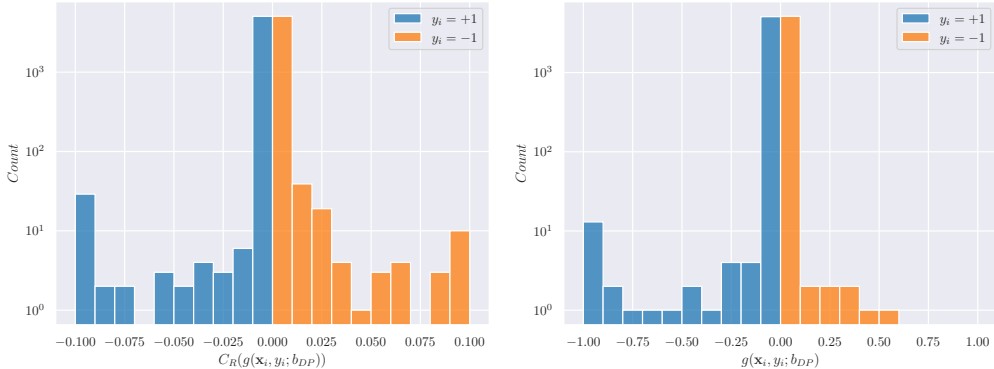

Figure 4: Distribution of gradient with clipping (left, which is balanced) and without clipping (right, which is unbalanced) of linear classifiers for (5), $K = 4, \sigma = 0.2, \theta = 1$, and clipping norm $R = 0.1$.

**Theorem 3.** *Fixing the attack magnitude $\gamma$, if data distribution in Equation (5) satisfies $\frac{K^2+1}{2K}\gamma < |\theta - \gamma| + |\theta|$, then whenever $b_\gamma < b_{\mathrm{DP}} < b_0$, we have*

$$\min_{f \ is \ linear} \mathcal{R}_\gamma(f) \equiv \mathcal{R}_\gamma(f_\gamma) < \mathcal{R}_\gamma(f_{\mathrm{DP}}) < \mathcal{R}_\gamma(f_0).$$

*Furthermore, any intercept $b$ with better natural accuracy than $b_{\mathrm{DP}}$ must have worse robust accuracy:*

$$\mathcal{R}_0(f) < \mathcal{R}_0(f_{\mathrm{DP}}) \Longrightarrow \mathcal{R}_\gamma(f) > \mathcal{R}_\gamma(f_{\mathrm{DP}}).$$

Theorem 3 shows that, under some conditions, DP models are more robust than natural models, and cannot be dominated in the Pareto optimal sense. That is, DP linear classifier can be Pareto optimal in terms of the robust and natural accuracy. We visualize the premise $b_\gamma < b_{\mathrm{DP}} < b_0$ in Figure 3 as well as in Figure 1, and the result $\mathcal{R}_\gamma(f_\gamma) < \mathcal{R}_\gamma(f_{\mathrm{DP}}) < \mathcal{R}_\gamma(f_0)$ in Figure 2.

We emphasize that, our results on the $l_\infty$ attacks is generally extendable to $l_2$ attacks. Put differently, we show that DP models can be adversarially robust and Pareto optimal under both $l_\infty$ and $l_2$ attacks[5].

**Corollary 1** (Extension to $l_2$ attacks). *All theorems hold for $l_2$ attacks by changing $\gamma \to \gamma/\sqrt{d}$.*

*Remark* 3. Theorem 2 gives sufficient and necessary condition for the DP classifier to be more robust than all classifiers at one attack magnitude $\gamma^*$; Theorem 3 gives sufficient but not necessary condition for the DP classifier to be more robust than one classifier (the natural one) at many attack magnitudes.

## 4 Training Private and Robust Neural Networks

In this section, we extend our investigation beyond the linear classifiers in Theorem 1, Theorem 2, and Theorem 3, and study the robustness of DP neural networks. We emphasize that several state-of-the-art (SOTA) advances are actually achieved by *linear classifiers* within the deep neural networks (Mehta et al., 2022; Tramer & Boneh, 2020), i.e. by finetuning only the last linear layer of Wide ResNet, SimCLR, and vision transformers. Therefore these advances fall in the same setting as our theorems, although the neural networks are non-linear models in general.

By experimenting with real datasets MNIST, CIFAR10 and CelebA, we corroborate our insights gained from theoretically analyzing the linear models and empirically show that the DP neural networks can be adversarially robust in practice (despite being much more challenging to analyze). We use one Nvidia GTX 1080Ti GPU and the Renyi privacy accountant to calculate the privacy loss.

---

[5]In DP deep learning, the clipping is on the gradient level under $l_2$ norm (see Footnote 1), regardless of the $l$ norm in adversarial attacks, which are on the sample level, i.e. on $\boldsymbol{x}_i$.

In summary, we have three key observations.

1. By selecting the hyper-parameters carefully, we can remain exactly the same level of DP but vastly stronger robustness. Especially, we visualize the distinct landscapes of robust accuracy and natural accuracy over $(R, \eta)$, which also depend on the choice of optimizers (see Appendix E).

2. We observe a privacy-accuracy-robustness tradeoff, showing that DP models is Pareto optimal (with or without pre-training), thus extending Theorem 3 to deep learning.

3. The robustness of DP models holds for general attacks, including single-step or multi-step (FGSM v.s. PGD), single method or ensemble (PGD v.s. APGD), and $l_\infty$ or $l_2$.

*Remark* 4. Our analysis also implies that DP models can be resilient to data poisoning attacks, since adversarial examples can serve as strong data poisons (Fowl et al., 2021). Such resilience is empirically observed in (Yang et al., 2022; Hong et al., 2020).

## 4.1 Hyper-parameters are keys to robustness

In DP deep learning, the training hyper-parameters can be divided into two categories: some are related to the privacy accounting, including the batch size $B$, the noise multiplier $\sigma_{\text{DP}}$, the number of iterations $T$; the others are only related to the optimization but not to the privacy, including the clipping norm $R$ and the learning rate $\eta$. That is, changing $(R, \eta)$ can influence the accuracy and the robustness without affecting the DP guarantee.

On one hand, $R$ has to be small to achieve SOTA natural accuracy. Large models such as ResNet and GPT2 are optimally trained at $R < 1$, even though the gradient's dimension is of hundreds of millions Kurakin et al. (2022); Li et al. (2021); Klause et al. (2022); Mehta et al. (2022). In fact, Bu et al. (2023b) adopts an infinitely small $R = 0^+$ to achieve SOTA results, essentially applying per-sample gradient normalization instead of the clipping.

On the other hand, DP training empirically benefits from large learning rate, usually 10 times larger than the non-DP training. This pattern is observed for DP-Adam (Li et al., 2021, Figure 4) and for DP-SGD (Kurakin et al., 2022) over text and image datasets.

Interestingly, we also observe such choice of $(R, \eta)$ performs strongly in the adversarial robustness context (though not the same hyper-parameters). By the ablation study in Figure 5 for CIFAR10 and in Appendix B for MNIST, Fashion MNIST and CelebA, it is clear that robust accuracy and natural accuracy have distinctively different landscapes over $(R, \eta)$. We observe that the optimal $(R, \eta)$ should be carefully selected along the diagonal ridge for DP-SGD to obtain high robust and high natural accuracy. Otherwise, even small deviation can lead to a sharp drop in the robustness, despite that the natural accuracy may remain similar (see upper right corner of 2D plots in Figure 5).

*Remark* 5. From Figure 5 (see also Figure 7, Figure 8 and Figure 9 in the appendix), it is empirically sufficient to set $R \approx 0^+$ and to only tune the learning rate $\eta$ for both robust and natural accuracy, when using DP optimizers such as DP-SGD, DP-Adam, DP-RMSprop and DP-Adagrad. This approach, termed as automatic clipping by Bu et al. (2023b), reduces the 2-dimension hyperparameter search to a much cheaper 1-dimension search (c.f. Table 1).

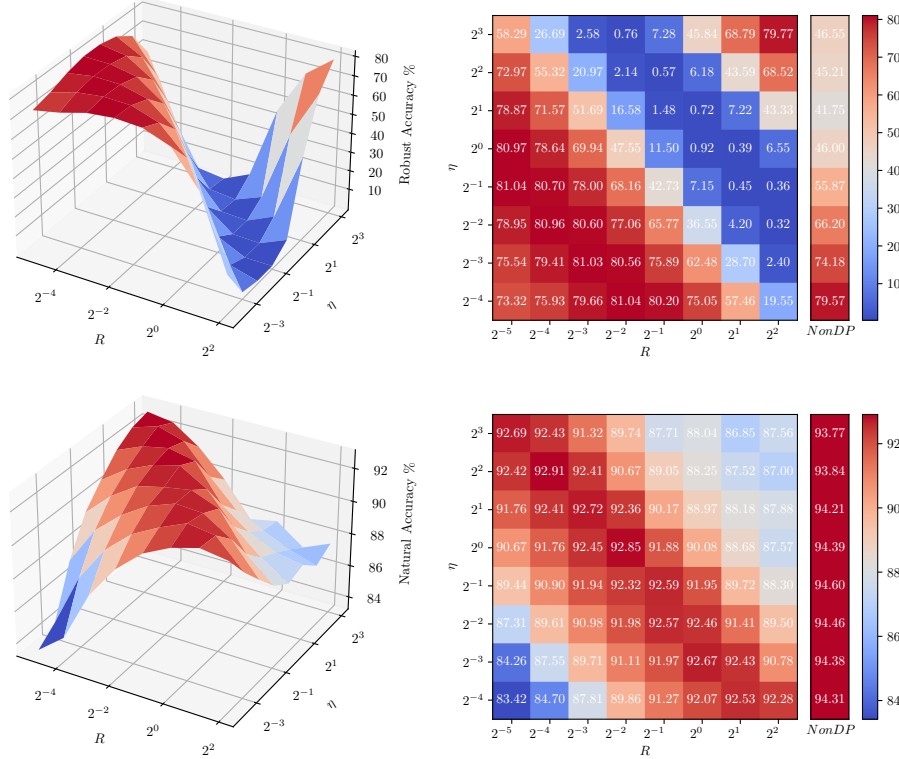

Figure 5: Robust and natural accuracy by $\eta$ and $R$ on CIFAR10. We use the same setting as in Tramer & Boneh (2020): pretraining SimCLR on ImageNet and then privately training using DP-SGD with momentum=0.9, under $(\epsilon, \delta) = (2, 1e-5)$ and attacked by 20 steps of $l_\infty(2/255)$ PGD.

Our ablation study demonstrates that the DP neural network, with 81.04% robust accuracy and 89.86% natural accuracy, can be more robust than the most robust of naturally trained networks (79.59% robust accuracy and 94.31% natural accuracy). If we trade some robustness for the natural accuracy, we can achieve the same level of robustness (80.20%) at 91.27% natural accuracy, thus closing the gap between the natural accuracy of DP and non-DP models without sacrificing the robustness.

While Figure 5 presents the result of a single attack magnitude, we further study the influence of hyper-parameters under different attack magnitudes, with and without the adversarial training. We illustrate on CIFAR10 the $l_\infty$ attack performance in Table 2 and the $l_2$ one in Table 3.

Table 2: Natural and robust accuracy of SimCLRv2 (Chen et al., 2020) and ResNet50 (Engstrom et al., 2019) on CIFAR10 under 20 steps $l_\infty$ PGD attack. Here cyan columns are adversarial training and white columns are natural training. Adversarial training and *robust* hyper-parameters are obtained by grid search over $\eta$ and $R$ against $l_\infty(2/255)$, and *natural* hyper-parameters are adopted from Tramer & Boneh (2020).

| | SimCLRv2 pre-trained on unlabelled ImageNet | | | | | | | | | | | | ResNet50 | |
|---|---|---|---|---|---|---|---|---|---|---|---|---|---|---|
| | DP | DP | DP | DP | DP | DP | DP | DP | DP | Non-DP | Non-DP | Non-DP | Non-DP | Non-DP |
| attack | $\epsilon = 2$ | $\epsilon = 2$ | $\epsilon = 2$ | $\epsilon = 4$ | $\epsilon = 4$ | $\epsilon = 4$ | $\epsilon = 8$ | $\epsilon = 8$ | $\epsilon = 8$ | $\epsilon = \infty$ | $\epsilon = \infty$ | $\epsilon = \infty$ | $\epsilon = \infty$ | $\epsilon = \infty$ |
| magnitude | adv $2/255$ | robust | accurate | adv $2/255$ | robust | accurate | adv $2/255$ | robust | accurate | adv $2/255$ | robust | accurate | adv $8/255$ | accurate |
| $\gamma = 0$ | 90.46% | 89.69% | 92.87% | 91.12% | 90.91% | 93.41% | 91.70% | 91.22% | 93.64% | 93.42% | 94.29% | 94.55% | 87.03% | 95.25% |
| $\gamma = 2/255$ | 83.83% | 81.05% | 33.21% | 85.28% | 82.53% | 57.80% | 86.12% | 83.02% | 68.90% | 89.07% | 79.79% | 59.56% | – | – |
| $\gamma = 4/255$ | 75.56% | 68.85% | 0.16% | 77.73% | 70.21% | 9.69% | 78.62% | 71.08% | 28.09% | 83.07% | 53.56% | 15.99% | – | – |
| $\gamma = 8/255$ | 53.61% | 39.63% | 0.00% | 56.90% | 38.39% | 0.00% | 57.84% | 39.28% | 0.01% | 66.99% | 8.14% | 0.00% | 53.49% | 0.00% |
| $\gamma = 16/255$ | 8.05% | 1.20% | 0.00% | 10.30% | 0.65% | 0.00% | 11.31% | 0.91% | 0.00% | 20.67% | 0.00% | 0.00% | 18.13% | 0.00% |

Table 3: Natural and robust accuracy of SimCLRv2 (Chen et al., 2020) and ResNet50 (Engstrom et al., 2019) on CIFAR10 under 20 steps $l_2$ PGD attack. Here cyan columns are adversarial training and white columns are natural training. Adversarial training and *robust* hyper-parameters are obtained by grid search over $\eta$ and $R$ against $l_2(0.25)$, and *natural* hyper-parameters are adopted from Tramer & Boneh (2020).

| | SimCLRv2 pre-trained on unlabelled ImageNet | | | | | | | | | | | | ResNet50 | |
| | DP | DP | DP | DP | DP | DP | DP | DP | DP | Non-DP | Non-DP | Non-DP | Non-DP | Non-DP |
| attack | $\epsilon=2$ | $\epsilon=2$ | $\epsilon=2$ | $\epsilon=4$ | $\epsilon=4$ | $\epsilon=4$ | $\epsilon=8$ | $\epsilon=8$ | $\epsilon=8$ | $\epsilon=\infty$ | $\epsilon=\infty$ | $\epsilon=\infty$ | $\epsilon=\infty$ | $\epsilon=\infty$ |
| magnitude | adv 0.25 | robust | accurate | adv 0.25 | robust | accurate | adv 0.25 | robust | accurate | adv 0.25 | robust | accurate | adv 0.5 | accurate |
| $\gamma=0$ | 91.07% | 89.69% | 92.87% | 91.19% | 90.91% | 93.41% | 91.87% | 91.22% | 93.64% | 93.88% | 94.29% | 94.55% | 90.83% | 95.25% |
| $\gamma=0.25$ | 82.69% | 82.12% | 59.91% | 83.69% | 83.35% | 74.10% | 84.27% | 83.77% | 79.03% | 85.20% | 82.91% | 72.63% | 82.34% | 8.66% |
| $\gamma=0.5$ | 70.54% | 71.99% | 12.76% | 72.42% | 72.79% | 40.97% | 72.00% | 73.08% | 54.53% | 71.22% | 63.32% | 35.95% | 70.17% | 0.28% |
| $\gamma=1.0$ | 38.57% | 46.30% | 9.49% | 42.97% | 44.46% | 8.97% | 40.17% | 44.65% | 9.68% | 39.74% | 33.34% | 0.98% | 40.47% | 0.00% |

We evaluate the robust and natural accuracy on the DP models in Tramer & Boneh (2020), considering two groups of hyper-parameters: the *natural* hyper-parameters reproduced from Tramer & Boneh (2020) that has highest natural accuracy, and the *robust* hyper-parameters from a grid search on $(R, \eta)$ for the highest robust accuracy. From Table 2 and Table 3, we see that even under the same privacy constraint (including the non-DP scenario), the robustness from different hyper-parameters can be fundamentally different. For example, DP SimCLR at $\epsilon = 2$ can be either very robust ($\approx 70\%$ accuracy at $\gamma = 4/255$) or not robust at all (0.16% accuracy). Consequently, our results may explain the mis-understanding of previous researches by the improper choice of the hyper-parameters.

Scrutinizing the natural training with robust hyper-parameters, we see that, across all $l_\infty$ attack magnitudes $\gamma = \{2/255, 4/255, 8/255, 16/255\}$ and $l_2$ ones $\gamma = \{0, 0.25, 0.5, 1.0\}$, DP SimCLR can be more robust than the non-DP SimCLR, in fact comparable to the adversarially trained ResNet50 that is benchmarked in Engstrom et al. (2019) and to the adversarially trained DP SimCLR. To be assured, we further demonstrate that our choice of small $R$ and large $\eta$ is consistently robust on Fashion MNIST, CIFAR10 and CelebA in Appendix C.

## 4.2 Pareto optimality on accuracy and robustness

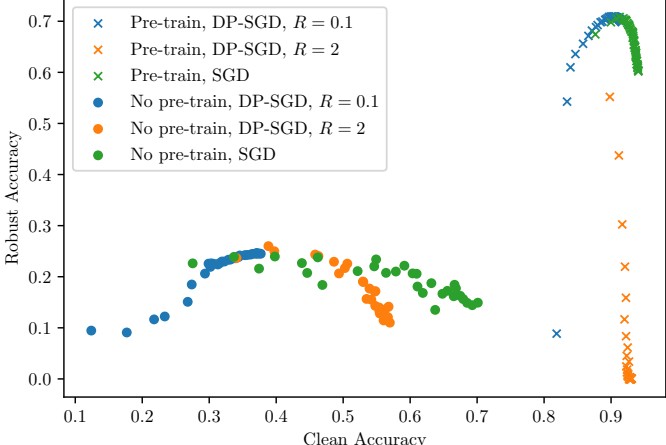

Figure 6: Robust and natural accuracy on CIFAR10 at different iterations. Dots are CNN from Papernot et al. (2020). Crosses are SimCLR from Tramer & Boneh (2020). See details in Appendix D.

In the standard non-DP regime, the tradeoff between the accuracy and the robustness is well-known (Engstrom et al., 2019). We extend the Pareto statement in Theorem 3 to DP deep learning, thus adding the privacy dimension into the privacy-accuracy-robustness tradeoff. In Figure 6, we show that two strong DP models on CIFAR10 (one pre-trained, the other not) achieve Pareto optimality with proper hyperparamters,

and thus cannot be dominated by any natural classifiers. This can be observed by the fact that no green cross (or dot) is to the top right of all blue crosses (or dots), meaning that any natural classifier may have better robustness or higher accuracy, but not both. Therefore, our observation supports the claim that DP neural networks can be Pareto optimal in terms of the robustness and the accuracy.

### 4.3 DP neural networks can be robust against general attacks

Following the claim in Section 4.1 that DP neural networks can be robust against $l_2$ and $l_\infty$ PGD attacks, we now demonstrate the transferability of DP neural networks' robustness against different attacks.

Table 4: Natural and robust accuracy of FGSM(Goodfellow et al., 2014), BIM(Kurakin et al., 2018), $PGD_\infty$(Madry et al., 2017), $APGD_\infty$(Croce & Hein, 2020), $PGD_2$(Madry et al., 2017) and $APGD_2$(Croce & Hein, 2020) on CIFAR10 under general adversarial attacks. Same model as Table 2 with the *robust* hyper-parameters. See detailed attack settings in Appendix D.

| | Natural | FGSM | BIM | $PGD_\infty$ | $APGD_\infty$ | $PGD_2$ | $APGD_2$ |
|---|---|---|---|---|---|---|---|
| DP , $\epsilon = 2$ | 89.86% | 69.73% | 68.85% | 68.85% | 68.85% | 72.10% | 71.97% |
| DP , $\epsilon = 4$ | 90.91% | 71.13% | 70.21% | 70.21% | 70.21% | 72.79% | 72.63% |
| DP , $\epsilon = 8$ | 91.22% | 71.88% | 71.08% | 71.08% | 71.06% | 73.08% | 72.99% |
| Non-DP | 94.29% | 56.24% | 53.56% | 53.56% | 53.56% | 63.32% | 62.93% |

This is interesting in the sense that DP mechanism does not intentionally defend against any adversarial attack, while the adversarial training (Goodfellow et al., 2014) usually specifically targets a particular attack, e.g. PGD attack is defended by PGD adversarial training. In Table 4, we attack on the robust models from Table 2, with the *robust* hyper-parameters. We consistently observe that DP models can be adversarially robust and more so than the non-DP ones on MNIST/Fashion MNIST/CelebA in Appendix C, if the hyper-parameters $(R, \eta)$ are set properly.

### 4.4 Large scale experiments on CelebA face datasets

We further validate our claims on CelebA (Liu et al., 2015), a public high-resolution ($178 \times 218$ pixels) image dataset, consisting of over 200,000 real human faces that are supposed to be protected against privacy risks. We train ResNet18 (He et al. (2016), 11 million parameters) and Vision Transformer (ViT, Dosovitskiy et al. (2020), 6 million parameters) with DP-RMSprop. Both models are implemented by Wightman (2019) and pretrained on ImageNet. The experiment can be reproduced using the DP vision codebase `Private Vision` by (Bu et al., 2022a).

Table 5: Natural and robust accuracy on CelebA with label 'Smiling', DP-RMSprop, under 20 steps $l_\infty$ PGD attack. Here the hyper-parameters have *not* been carefully searched for the best robustness. See details in Appendix D.

| | ResNet18 | | | | ViT | | | |
|---|---|---|---|---|---|---|---|---|
| attack | DP | DP | DP | Non-DP | DP | DP | DP | Non-DP |
| magnitude | $\epsilon = 2$ | $\epsilon = 4$ | $\epsilon = 8$ | $\epsilon = \infty$ | $\epsilon = 2$ | $\epsilon = 4$ | $\epsilon = 8$ | $\epsilon = \infty$ |
| $\gamma = 0$ | 80.10% | 85.10% | 88.48% | 91.91% | 92.30% | 92.33% | 92.09% | 92.87% |
| $\gamma = 2/255$ | 1.26% | 0.47% | 1.03% | 1.19% | 1.42% | 2.02% | 10.35% | 0.08% |
| $\gamma = 4/255$ | 0.01% | 0.01% | 0.00% | 0.00% | 0.00% | 0.00% | 0.00% | 0.00% |
| $\gamma = 8/255$ | 0.00% | 0.00% | 0.00% | 0.00% | 0.00% | 0.00% | 0.00% | 0.00% |
| $\gamma = 16/255$ | 0.00% | 0.00% | 0.00% | 0.00% | 0.00% | 0.00% | 0.00% | 0.00% |

Table 6: Natural and robust accuracy of FGSM(Goodfellow et al., 2014), BIM(Kurakin et al., 2018), PGD$_\infty$(Madry et al., 2017), APGD$_\infty$(Croce & Hein, 2020), PGD$_2$(Madry et al., 2017) and APGD$_2$(Croce & Hein, 2020) on CelebA with label 'Smiling' under general adversarial attacks. Same ResNet18 as Table 5. See detailed attack settings in Appendix D.

| | Natural | FGSM | BIM | PGD$_\infty$ | APGD$_\infty$ | PGD$_2$ | APGD$_2$ |
|---|---|---|---|---|---|---|---|
| DP , $\epsilon = 2$ | 80.10% | 24.47% | 1.24% | 1.26% | 1.18% | 47.02% | 46.25% |
| DP , $\epsilon = 4$ | 85.10% | 24.40% | 0.45% | 0.47% | 0.41% | 56.92% | 56.09% |
| DP , $\epsilon = 8$ | 88.48% | 29.32% | 0.97% | 1.03% | 0.41% | 57.40% | 56.69% |
| Non-DP | 91.91% | 22.94% | 1.09% | 1.19% | 0.68% | 66.89% | 66.13% |

In Table 5 and Table 6, we observe that DP ResNet18 and ViT are almost as adversarially robust as their non-DP counterparts, if not more robust. These observations are consistent with those of simpler models on tiny images (c.f. Table 2 and Table 4). We note that unlike the linear classifiers on CIFAR10 in Table 2, training all layers on CelebA in Table 5 are much more vulnerable even at $\gamma = 2/255$.

## 5    Discussion

Through the lens of theoretical analysis and extensive experiments, we have shown that differentially private models can be adversarially robust and sometimes even more robust than the naturally trained models. Moreover, DP models can be Pareto optimal in the sense that a more accurate natural model must be less robust (see Theorem 3 and Figure 6). Our conclusion holds for various attacks with different magnitudes, from linear models to large vision models, from grey-scale images to real face datasets, and from SGD to adaptive optimizers. We not only are the first to reveal this possibility of achieving privacy and robustness simultaneously, but also are the first to offer practical guidelines for such important goal (see Section 4). To be concrete, we demonstrate that hyper-parameters – clipping norm $R$ and learning rate $\eta$ – exert a lot of influence on the robustness and accuracy, while remaining equally private. We hope that our insights will encourage the practitioners to adopt techniques that protect the privacy and robustness in real-world applications.

For future directions, a more thorough study of private and robustness learning is desirable, by extending to language models, recommendation systems, and so on. We believe a new analysis when all parameters are trainable will be challenging but enlightening. Especially, given that larger models are empirically more accurate under the fixed privacy budget, it would be interesting to understand whether the robustness also improves, or at least persists, with larger model sizes. Another direction is to further investigate the adversarial training with DP optimizers, whose performance may go beyond our Pareto frontier (of the robustness and the accuracy) that is based on the natural training.

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

## A   Proofs

*Proof of Theorem 1.* [Extended from Xu et al. (2021)] By Xu et al. (2021, Lemma 2), according to the data symmetry in (5), the optimal linear classifier has the form

$$1, \cdots, 1, b_\gamma = \arg\min_{\mathbf{w}, b} \mathcal{R}_\gamma(f(\cdot; \mathbf{w}, b)).$$

Recall that (8) proves that for such linear classifier, the robust error is

$$\mathcal{R}_\gamma(f) = \frac{1}{2}\Phi\left(-\frac{\sqrt{d}(\theta - \gamma)}{\sigma} + \frac{1}{\sqrt{d}\sigma} \cdot b\right) + \frac{1}{2}\Phi\left(-\frac{\sqrt{d}(\theta - \gamma)}{K\sigma} - \frac{1}{K\sqrt{d}\sigma} \cdot b\right).$$

where $\Phi$ is the cumulative distribution function of standard normal.

The optimal $b_\gamma$ to minimize $\mathcal{R}_\gamma(f)$ is achieved at the point that $\frac{\partial \mathcal{R}_\gamma(f)}{\partial b} = 0$. Thus, $b_\gamma$ satisfies:

$$\phi\left(-\frac{\sqrt{d}(\theta - \gamma)}{\sigma} + \frac{b_\gamma}{\sqrt{d}\sigma}\right) \cdot \frac{1}{\sqrt{d}\sigma} - \phi\left(-\frac{\sqrt{d}(\theta - \gamma)}{K\sigma} - \frac{b_\gamma}{K\sqrt{d}\sigma}\right) \cdot \frac{1}{K\sqrt{d}\sigma} = 0$$

where $\phi$ is the probability density function of standard normal. This equals to

$$\phi\left(-\frac{\sqrt{d}(\theta - \gamma)}{\sigma} + \frac{b_\gamma}{\sqrt{d}\sigma}\right) = \phi\left(-\frac{\sqrt{d}(\theta - \gamma)}{K\sigma} - \frac{b_\gamma}{K\sqrt{d}\sigma}\right) / K$$

and

$$K = \phi\left(-\frac{\sqrt{d}(\theta - \gamma)}{K\sigma} - \frac{b_\gamma}{K\sqrt{d}\sigma}\right) / \phi\left(-\frac{\sqrt{d}(\theta - \gamma)}{\sigma} + \frac{b_\gamma}{\sqrt{d}\sigma}\right)$$

$$= e^{-\frac{1}{2}\left[\left(-\frac{\sqrt{d}(\theta-\gamma)}{K\sigma} - \frac{b_\gamma}{K\sqrt{d}\sigma}\right)^2 - \left(-\frac{\sqrt{d}(\theta-\gamma)}{\sigma} + \frac{b_\gamma}{\sqrt{d}\sigma}\right)^2\right]}$$

It is not hard to see

$$-2\log K = \left(-\frac{\sqrt{d}(\theta - \gamma)}{K\sigma} - \frac{b_\gamma}{K\sqrt{d}\sigma}\right)^2 - \left(-\frac{\sqrt{d}(\theta - \gamma)}{\sigma} + \frac{b_\gamma}{\sqrt{d}\sigma}\right)^2$$

which re-arranges to a quadratic equation

$$b_\gamma^2 \frac{1}{d\sigma^2}\left(1 - \frac{1}{K^2}\right) - b_\gamma \frac{2(\theta - \gamma)}{\sigma^2}\left(1 + \frac{1}{K^2}\right) + \frac{d(\theta - \gamma)^2}{\sigma^2}\left(1 - \frac{1}{K^2}\right) = 2\log K.$$

The solution is therefore explicit as

$$b_\gamma = \frac{K^2 + 1}{K^2 - 1}d(\theta - \gamma) - K\sqrt{\frac{4d^2(\theta - \gamma)^2}{(K^2 - 1)^2} + d\sigma^2 q(K)},$$

where $q(K) = \frac{2\log K}{K^2 - 1}$ which is a positive constant and only depends on $K$. By incorporating $b_\gamma$ into (8), we can get the optimal robust error $\mathcal{R}_\gamma(f_\gamma)$:

$$\mathcal{R}_\gamma(f_\gamma) = \frac{1}{2}\Phi\left(B(K, \gamma) - K\sqrt{B(K, \gamma)^2 + q(K)}\right) + \frac{1}{2}\Phi\left(-KB(K, \gamma) + \sqrt{B(K, \gamma)^2 + q(K)}\right),$$

where $B(K, \gamma) = \frac{2}{K^2 - 1}\frac{\sqrt{d}(\theta - \gamma)}{\sigma}$. □

*Proof of Theorem 3.* We denote the two roots of $\frac{\partial \mathcal{R}_\gamma(f(b))}{\partial b} = 0$ as $b_\gamma^+$ and $b_\gamma^-$. Here $b_\gamma \equiv b_\gamma^-$. Clearly $\mathcal{R}_\gamma(b)$ is increasing in $(b_\gamma^-, b_\gamma^+)$. We hope to show $b_0 \in (b_\gamma^-, b_\gamma^+) \forall \gamma > 0$, so that $\mathcal{R}_\gamma(b)$ is also increasing in $(b_\gamma^-, b_0)$.

Note their Equation (17)

$$\mathcal{R}_\gamma(b) = \frac{1}{2}\Phi(-\frac{\sqrt{d}(\theta - \gamma)}{\sigma} + \frac{1}{\sqrt{d}\sigma}b) + \frac{1}{2}\Phi(-\frac{\sqrt{d}(\theta - \gamma)}{K\sigma} - \frac{1}{K\sqrt{d}\sigma}b)$$

Taking derivative w.r.t. $b$

$$\frac{\partial \mathcal{R}_\gamma(b)}{\partial b} = \frac{1}{2\sqrt{d}\sigma}\phi(-\frac{\sqrt{d}(\theta - \gamma)}{\sigma} + \frac{1}{\sqrt{d}\sigma}b) - \frac{1}{2K\sqrt{d}\sigma}\phi(-\frac{\sqrt{d}(\theta - \gamma)}{K\sigma} - \frac{1}{K\sqrt{d}\sigma}b)$$

Setting this derivative to 0:

$$0 = K\phi(-\frac{\sqrt{d}(\theta - \gamma)}{\sigma} + \frac{1}{\sqrt{d}\sigma}b) - \phi(-\frac{\sqrt{d}(\theta - \gamma)}{K\sigma} - \frac{1}{K\sqrt{d}\sigma}b)$$

which means

$$\frac{\phi(-\frac{\sqrt{d}(\theta-\gamma)}{K\sigma} - \frac{1}{K\sqrt{d}\sigma}b)}{\phi(-\frac{\sqrt{d}(\theta-\gamma)}{\sigma} + \frac{1}{\sqrt{d}\sigma}b)} = K$$

Using the standard normal density $\phi(u) = e^{-u^2/2}$ and $\frac{\phi(u)}{\phi(v)} = e^{(v^2 - u^2)/2}$, we have

$$(-\frac{\sqrt{d}(\theta - \gamma)}{\sigma} + \frac{1}{\sqrt{d}\sigma}b)^2 - (-\frac{\sqrt{d}(\theta - \gamma)}{K\sigma} - \frac{1}{K\sqrt{d}\sigma}b)^2 = 2\log K$$

$$\Longrightarrow K^2(-d(\theta - \gamma) + b)^2 - (-d(\theta - \gamma) - b)^2 = 2d\sigma^2 K^2 \log K$$

$$\Longrightarrow (K^2 - 1)b^2 - 2d(\theta - \gamma)(K^2 + 1)b + d^2(\theta - \gamma)^2(K^2 - 1) - 2d\sigma^2 K^2 \log K = 0$$

By $x = -\frac{b}{2a} \pm \frac{\sqrt{b^2 - 4ac}}{2a} = -\frac{b}{2a} \pm \sqrt{(\frac{b}{2a})^2 - \frac{c}{a}}$, we know

$$b_\gamma^\pm = \frac{K^2 + 1}{K^2 - 1}d(\theta - \gamma) \pm \sqrt{(\frac{K^2 + 1}{K^2 - 1}d(\theta - \gamma))^2 - d^2(\theta - \gamma)^2 + K^2 d\sigma^2 q(K)}$$

$$= \frac{K^2 + 1}{K^2 - 1}d(\theta - \gamma) \pm K\sqrt{\frac{4d^2(\theta - \gamma)^2}{(K^2 - 1)^2} + d\sigma^2 q(K)}$$

We now derive the sufficient condition that $b_0 < b_\gamma^+$:

$$\frac{K^2 + 1}{K^2 - 1}d(\theta) - K\sqrt{\frac{4d^2(\theta)^2}{(K^2 - 1)^2} + d\sigma^2 q(K)} < \frac{K^2 + 1}{K^2 - 1}d(\theta - \gamma) + K\sqrt{\frac{4d^2(\theta - \gamma)^2}{(K^2 - 1)^2} + d\sigma^2 q(K)}.$$

This is equivalent to

$$\frac{K^2 + 1}{K^2 - 1}d\gamma < K\left(\sqrt{\frac{4d^2(\theta - \gamma)^2}{(K^2 - 1)^2} + d\sigma^2 q(K)} + \sqrt{\frac{4d^2\theta^2}{(K^2 - 1)^2} + d\sigma^2 q(K)}\right).$$

Therefore, it suffices to have

$$\frac{K^2 + 1}{2K}\gamma < |\theta - \gamma| + |\theta|$$

Finally, it is easy to see the Pareto statement $\mathcal{R}_0(f) < \mathcal{R}_0(f_{\text{DP}}) \longrightarrow \mathcal{R}_\gamma(f) > \mathcal{R}_\gamma(f_{\text{DP}})$. A necessary but not sufficient condition for $\mathcal{R}_0(f) < \mathcal{R}_0(f_{\text{DP}})$ given that $b_0 > b_{\text{DP}}$ is $b > b_{\text{DP}}$, since $b_0$ is a minimizer which means $\mathcal{R}_0$ is decreasing on the interval $(-\infty, b_0)$. Similarly, $\mathcal{R}_\gamma$ is increasing on the right of $b_\gamma$ and thus $b$ has higher robust error. $\qquad\square$

*Proof of Corollary 1.* We can characterize the robust errors based on $l_2$ attacks in a similar fashion to (8). We notice that

$$\mathcal{R}_\gamma(f) = \mathbb{P}(\exists \|\boldsymbol{p}\|_2 \le \epsilon \text{ s.t. } f(\boldsymbol{x} + \boldsymbol{p}) \ne y) = \max_{\|\boldsymbol{p}\|_2 \le \gamma} \mathbb{P}(f(\boldsymbol{x} + \boldsymbol{p}) \ne y)$$

$$= \frac{1}{2}\mathbb{P}(f(\boldsymbol{x} + \boldsymbol{\gamma}_d/\sqrt{d}) \ne -1 \mid y = -1) + \frac{1}{2}\mathbb{P}(f(\boldsymbol{x} - \boldsymbol{\gamma}_d/\sqrt{d}) \ne +1 \mid y = +1)$$

In short, the same analysis is in place except $\gamma \to \gamma/\sqrt{d}$ when we switch from $l_\infty$ to $l_2$ attacks. $\square$

# B  Ablation Studies

## B.1  CelebA

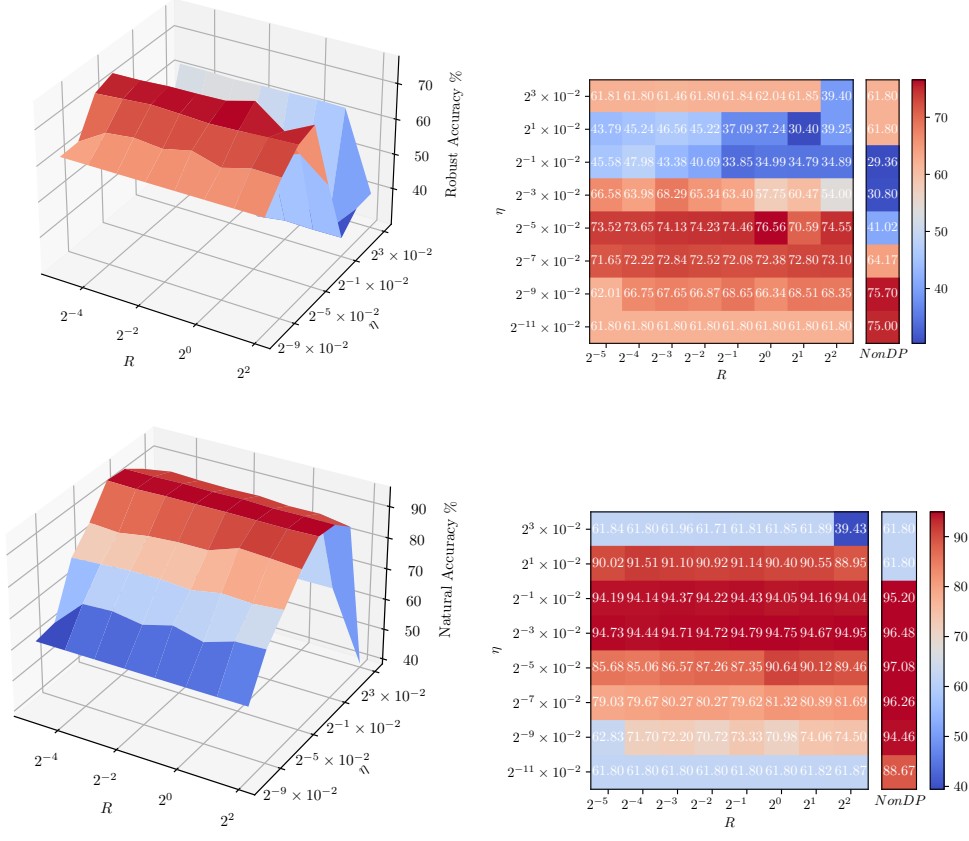

Figure 7: Robust and natural accuracy of $\eta$ and $R$ on CelebA with label 'Male'. We train a 2-layer CNN using DP-Adam and attack by $l_\infty(2/255)$ PGD attack. Same as in Figure 13. Here $\epsilon = 2$, batch size = 512, epochs = 10.

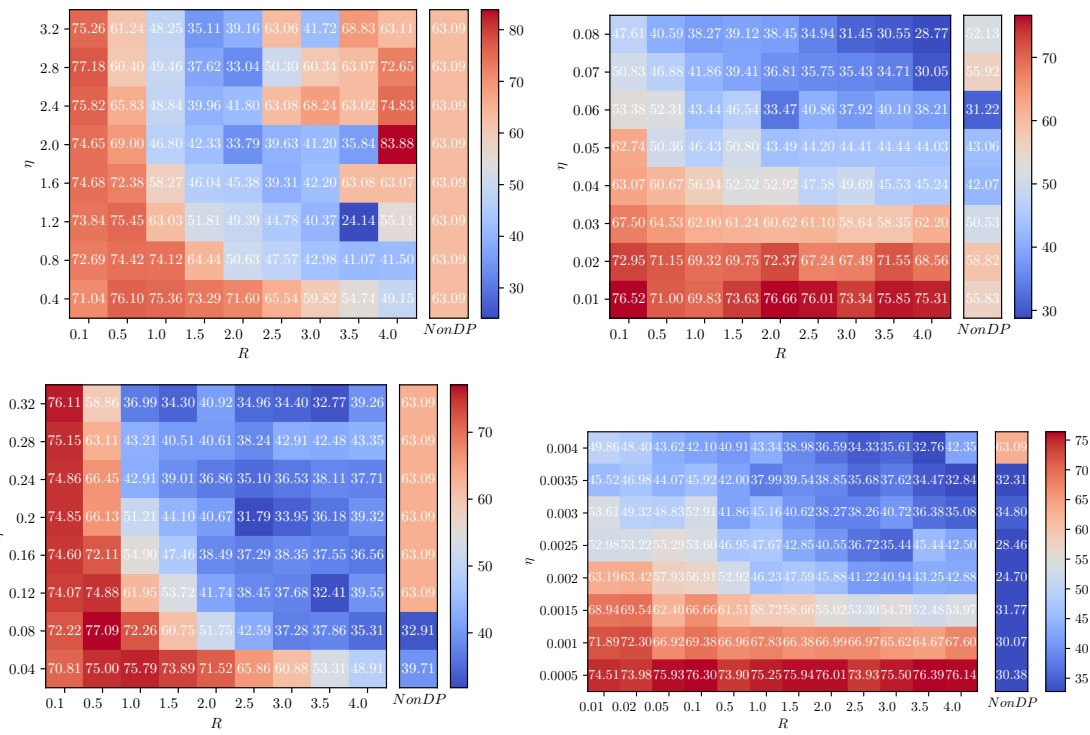

Figure 8: Robust accuracy of CelebA with label 'Male' under different optimizer, trained with a 2-layer CNN and attacked by $l_\infty(2/255)$ PGD attack. Top left: SGD. Top right: Adagrad. Bottom left: SGD momentum. Bottom right: Adam. Here $\epsilon = 2$, batch size $= 512$, epochs $= 10$.

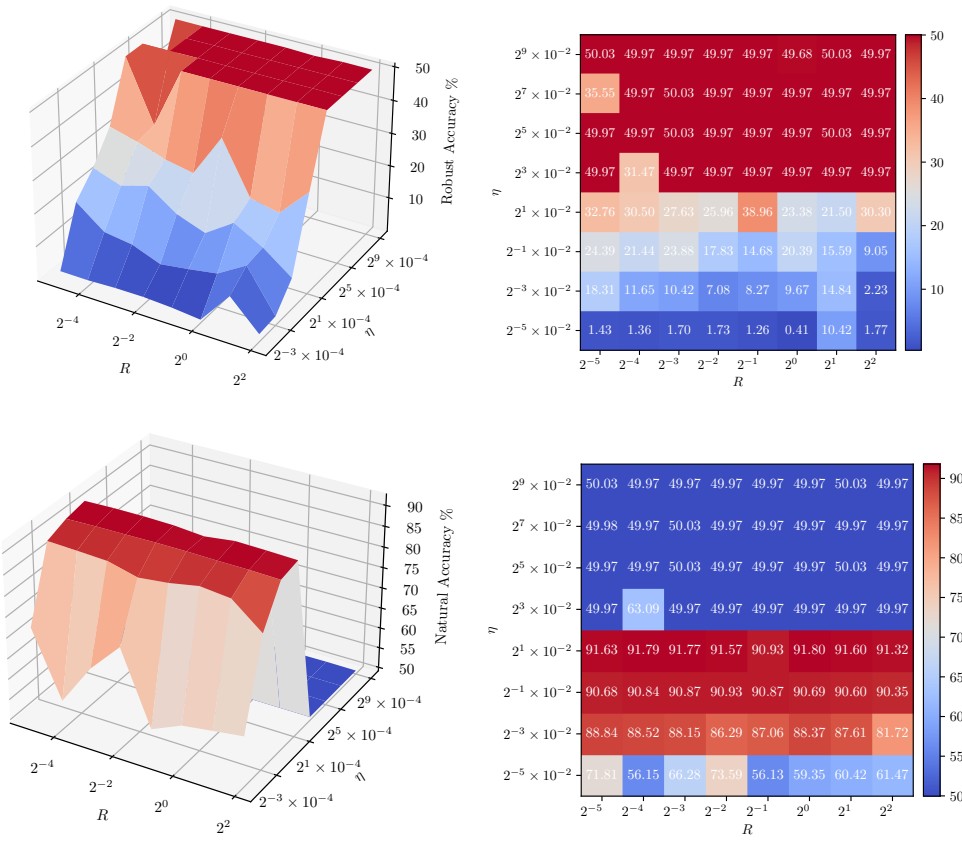

Figure 9: Robust and natural accuracy of $\eta$ and $R$ on CelebA with label 'Smiling'. We train ViT-tiny using DP-RMSprop and attack by $l_\infty(2/255)$ PGD attack. Here $\epsilon = 2$, batch size $= 1024$, epoch $= 1$.

## B.2 CIFAR10

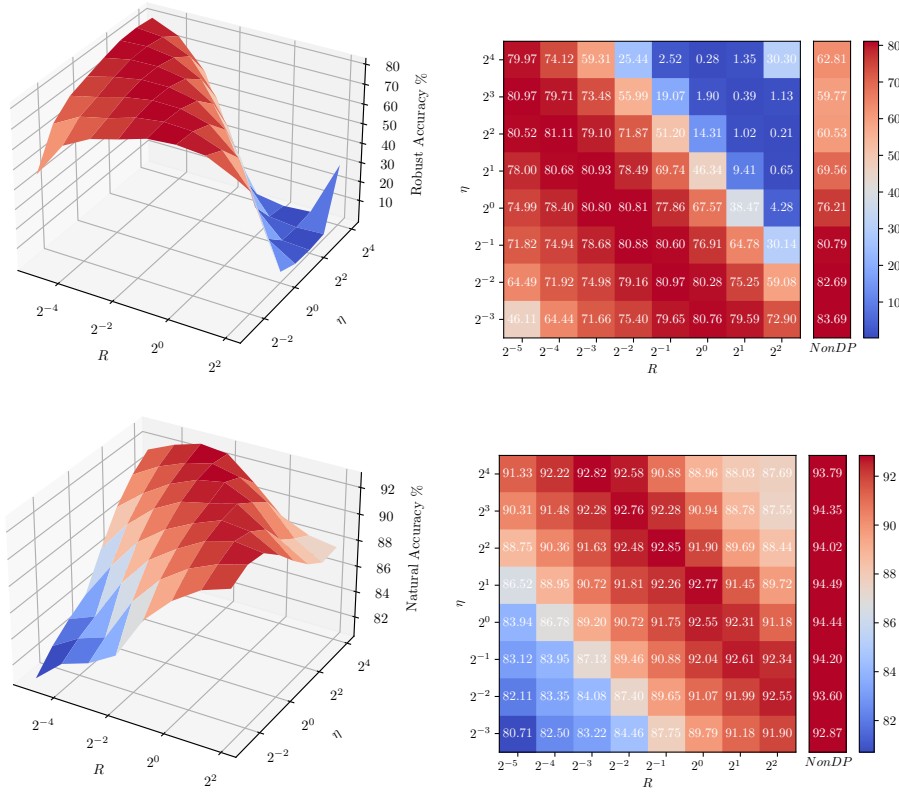

Figure 10: Robust and clean accuracy of $\eta$ and $R$ on CIFAR10, transferred from SimCLRv2 pre-trained on unlabelled ImageNet. We use DP-SGD and attack by $l_\infty(2/255)$ PGD attack. Here $\epsilon = 2$, batch size = 1024, epochs = 50.

## B.3 MNIST

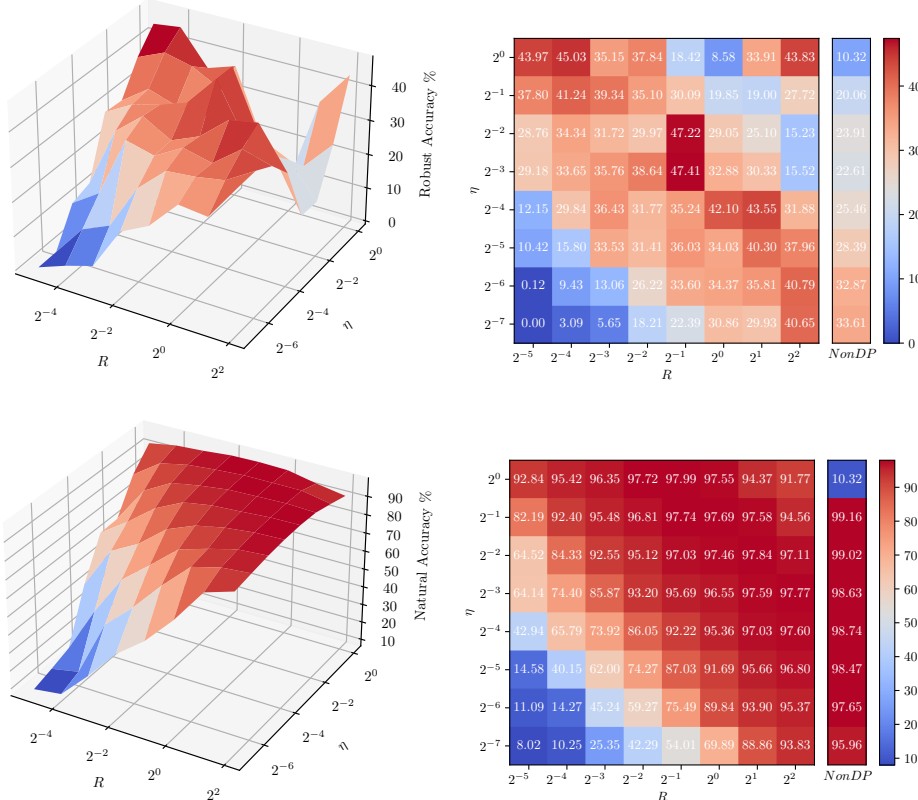

Figure 11: Robust and clean accuracy of $\eta$ and $R$ on MNIST. We train the CNN from Tramer & Boneh (2020) using DP-SGD and attack by $l_\infty(32/255)$ PGD attack. Here $\epsilon = 2$, batch size = 512, epochs = 40.

## B.4 Fashion MNIST

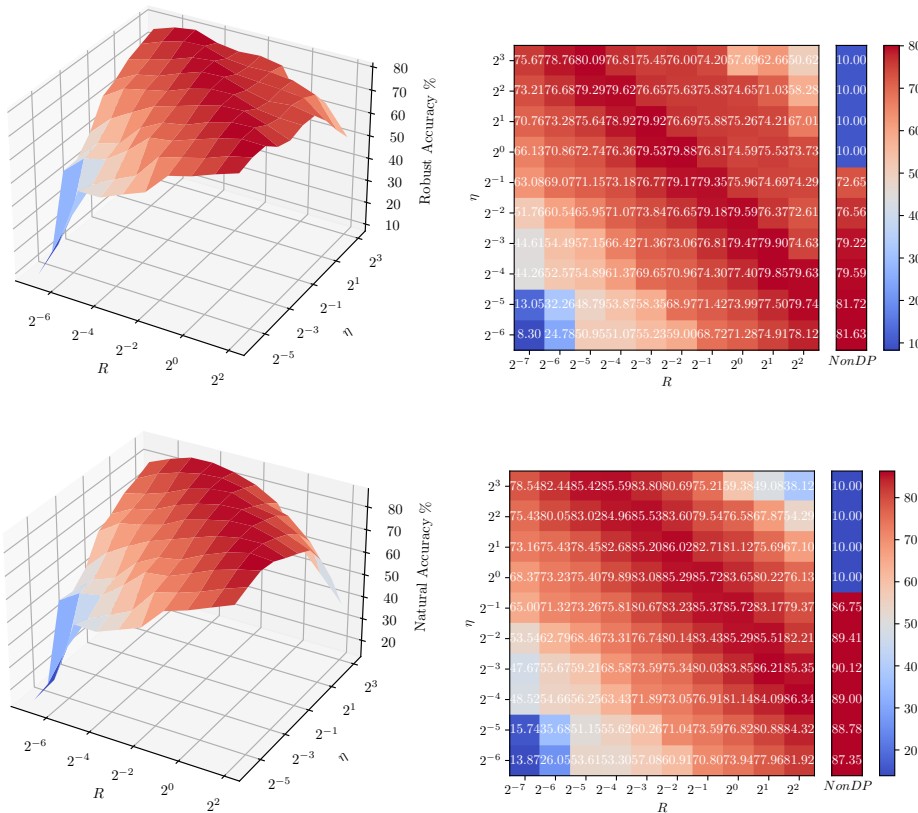

Figure 12: Robust and clean accuracy of $\eta$ and $R$ on Fashion MNIST. We train the CNN from Tramer & Boneh (2020) using DP-SGD and attack by $l_\infty(2/255)$ PGD attack. Here $\epsilon = 2$, batch size = 2048, epochs = 40.

## C   More tables

| | Natural | FGSM | BIM | $PGD_\infty$ | $APGD_\infty$ | $PGD_2$ | $APGD_2$ |
|---|---|---|---|---|---|---|---|
| Non-DP | 94.55% | 18.71% | 15.97% | 15.96% | 16.04% | 35.95% | 35.89% |
| DP , $\epsilon = 2$ | 92.73% | 10.35% | 0.03% | 0.03% | 0.03% | 12.76% | 12.68% |
| DP , $\epsilon = 4$ | 93.49% | 30.10% | 9.10% | 9.09% | 9.12% | 40.97% | 41.01% |
| DP , $\epsilon = 8$ | 93.74% | 31.86% | 28.08% | 28.09% | 28.09% | 54.53% | 54.54% |

Table 7: Natural and robust accuracy of models transferred from unlabelled ImageNet pre-trained SIMCLRv2 on CIFAR10 under general adversarial attacks with $\gamma_\infty = 4/255$ and $\gamma_2 = 0.5$. Attack steps are 20 if applicable. Model hyper-parameters are directly adopted from Tramer & Boneh (2020) for highest natural accuracy. DP models are trained using DP-SGD, $R = 0.1$, $\eta_{DP} = 4$, momentum = 0.9, batch size = 1024. Non-DP models are trained using SGD with the same hyper-parameters except $\eta_{non-DP} = 0.4$.

| | Non-DP | DP | DP | DP |
|---|---|---|---|---|
| attack magnitude | $\epsilon = \infty$ | $\epsilon = 2$ | $\epsilon = 4$ | $\epsilon = 8$ |
| $\gamma = 0.0$ | 99.24% | 98.01% | 98.32% | 98.50% |
| $\gamma = 0.25$ | 97.57% | 95.29% | 95.94% | 96.65% |
| $\gamma = 0.5$ | 93.32% | 90.28% | 91.71% | 92.97% |
| $\gamma = 1.0$ | 66.58% | 63.95% | 73.32% | 77.08% |
| $\gamma = 2.0$ | 36.28% | 39.88% | 51.48% | 52.74% |

Table 8: Robust accuracy on MNIST under 20 steps $l_2$ PGD attack. Model hyper-parameters are directly adopted from Tramer & Boneh (2020) for highest natural accuracy. DP models are trained using DP-SGD, $R = 0.1$, $\eta_{DP} = 0.5$, momentum = 0.9, batch size = 512. Non-DP models are trained using SGD with the same hyper-parameters except $\eta_{non-DP} = 0.05$.

| | Non-DP | DP | DP | DP |
|---|---|---|---|---|
| attack magnitude | $\epsilon = \infty$ | $\epsilon = 2$ | $\epsilon = 4$ | $\epsilon = 8$ |
| $\gamma = 0.0$ | 99.24% | 98.01% | 98.32% | 98.50% |
| $\gamma = 2/255$ | 98.73% | 97.12% | 97.43% | 97.84% |
| $\gamma = 4/255$ | 97.88% | 95.78% | 96.32% | 97.13% |
| $\gamma = 8/255$ | 95.32% | 92.31% | 93.51% | 94.74% |
| $\gamma = 16/255$ | 82.06% | 77.67% | 80.28% | 85.82% |

Table 9: Robust accuracy on MNIST under 20 steps $l_\infty$ PGD attack. Model hyper-parameters are directly adopted from Tramer & Boneh (2020) for highest natural accuracy. DP models are trained using DP-SGD, $R = 0.1$, $\eta_{DP} = 0.5$, momentum = 0.9, batch size = 512. Non-DP models are trained using SGD with the same hyper-parameters except $\eta_{non-DP} = 0.05$.

|  | Natural | FGSM | BIM | $PGD_\infty$ | $APGD_\infty$ | $PGD_2$ | $APGD_2$ |
|---|---|---|---|---|---|---|---|
| Non-DP | 99.24% | 97.92% | 97.88% | 97.88% | 97.77% | 93.32% | 93.27% |
| DP , $\epsilon = 2$ | 98.01% | 95.89% | 95.80% | 95.79% | 95.63% | 90.28% | 90.15% |
| DP , $\epsilon = 4$ | 98.32% | 96.45% | 96.32% | 96.33% | 96.27% | 91.71% | 91.68% |
| DP , $\epsilon = 8$ | 98.50% | 97.19% | 97.15% | 97.15% | 97.06% | 92.97% | 92.94% |

Table 10: Natural and robust accuracy of CNN models on MNIST under general adversarial attacks with $\gamma_\infty = 4/255$ and $\gamma_2 = 0.5$. Attack steps are 20 if applicable. Model hyper-parameters are directly adopted from Tramer & Boneh (2020) for highest natural accuracy. DP models are trained using DP-SGD, $R = 0.1$, $\eta_{DP} = 0.5$, momentum = 0.9, batch size = 512. Non-DP models are trained using SGD with the same hyper-parameters except $\eta_{non-DP} = 0.05$.

|  | Non-DP | DP | DP | DP |
|---|---|---|---|---|
| attack magnitude | $\epsilon = \infty$ | $\epsilon = 2$ | $\epsilon = 4$ | $\epsilon = 8$ |
| $\gamma = 0.0$ | 89.75% | 85.95% | 86.60% | 86.74% |
| $\gamma = 0.25$ | 57.37% | 69.24% | 72.93% | 75.35% |
| $\gamma = 0.5$ | 25.21% | 46.09% | 54.30% | 59.23% |
| $\gamma = 1.0$ | 7.87% | 16.77% | 25.95% | 29.08% |
| $\gamma = 2.0$ | 7.47% | 11.77% | 16.85% | 17.00% |

Table 11: Robust accuracy on Fashion MNIST under 20 steps $l_2$ PGD attack. Model hyper-parameters are directly adopted from Tramer & Boneh (2020) for highest natural accuracy. DP models are trained using DP-SGD, $R = 0.1$, $\eta_{DP} = 4$, momentum = 0.9, batch size = 2048. Non-DP models are trained using SGD with the same hyper-parameters except $\eta_{non-DP} = 0.4$.

|  | Non-DP | DP | DP | DP |
|---|---|---|---|---|
| attack magnitude | $\epsilon = \infty$ | $\epsilon = 2$ | $\epsilon = 4$ | $\epsilon = 8$ |
| $\gamma = 0.0$ | 89.75% | 85.95% | 86.60% | 86.74% |
| $\gamma = 2/255$ | 76.19% | 78.29% | 79.84% | 81.47% |
| $\gamma = 4/255$ | 64.46% | 69.75% | 72.60% | 74.72% |
| $\gamma = 8/255$ | 47.24% | 54.62% | 57.87% | 60.52% |
| $\gamma = 16/255$ | 23.26% | 28.51% | 31.68% | 30.90% |

Table 12: Robust accuracy on Fashion MNIST under 20 steps $l_\infty$ PGD attack. Model hyper-parameters are directly adopted from Tramer & Boneh (2020) for highest natural accuracy. DP models are trained using DP-SGD, $R = 0.1$, $\eta_{DP} = 4$, momentum = 0.9, batch size = 2048. Non-DP models are trained using SGD with the same hyper-parameters except $\eta_{non-DP} = 0.4$.

|  | Natural | FGSM | BIM | $PGD_\infty$ | $APGD_\infty$ | $PGD_2$ | $APGD_2$ |
|---|---|---|---|---|---|---|---|
| Non-DP | 89.75% | 70.41% | 64.56% | 64.44% | 53.41% | 25.21% | 23.13% |
| DP , $\epsilon = 2$ | 85.95% | 72.11% | 69.76% | 69.71% | 67.13% | 46.09% | 45.41% |
| DP , $\epsilon = 4$ | 86.60% | 73.67% | 72.68% | 72.69% | 70.84% | 54.30% | 53.92% |
| DP , $\epsilon = 8$ | 86.74% | 75.45% | 74.75% | 74.74% | 73.71% | 59.23% | 58.98% |

Table 13: Natural and robust accuracy of CNN models on Fashion MNIST under general adversarial attacks with $\gamma_\infty = 4/255$ and $\gamma_2 = 0.5$. Attack steps are 20 if applicable. Model hyper-parameters are directly adopted from Tramer & Boneh (2020) for highest natural accuracy. DP models are trained using DP-SGD, $R = 0.1$, $\eta_{DP} = 4$, momentum = 0.9, batch size = 2048. Non-DP models are trained using SGD with the same hyper-parameters except $\eta_{non-DP} = 0.4$.

|  | Natural | FGSM | BIM | $PGD_\infty$ | $APGD_\infty$ | $PGD_2$ | $APGD_2$ |
|---|---|---|---|---|---|---|---|
| Non-DP | 94.29% | 14.48% | 12.02% | 12.00% | 12.03% | 31.36% | 31.28% |
| DP , $\epsilon = 2$ | 92.73% | 15.70% | 1.59% | 1.61% | 1.62% | 28.05% | 28.06% |
| DP , $\epsilon = 4$ | 93.49% | 30.89% | 5.23% | 5.27% | 5.25% | 35.96% | 35.98% |
| DP , $\epsilon = 8$ | 93.74% | 9.66% | 4.30% | 4.29% | 4.31% | 33.21% | 33.23% |

Table 14: Natural and robust accuracy of models transferred from unlabelled ImageNet pre-trained SIM-CLRv2 on CIFAR10 under general adversarial attacks with $\gamma_\infty = 4/255$ and $\gamma_2 = 0.5$. Attack steps are 20 if applicable. Model in each row is the most accurate model obtained by simple grid search: Non-DP: $\eta = 0.5$; $DP_{\epsilon=2}$: $\eta = 1.0, R = 0.25$; $DP_{\epsilon=4}$: $\eta = 8, R = 0.0625$, $DP_{\epsilon=8}$: $\eta = 0.5, R = 1.0$. All models are trained using SGD or DP-SGD, momentum $= 0.9$ and batch size $= 1024$.

## D  Hyper-parameter setup

In Table 2, SimCLRv2 models are pre-trained on unlabelled ImageNet and fine-tuned on CIFAR10. *Natural* models are directly adopted from Tramer & Boneh (2020) for highest natural accuracy, where optimizer is DP-SGD and SGD, $R = 0.1$, $\eta_{DP} = 4$, $\eta_{non-DP} = 0.4$, momentum $= 0.9$, batch size $= 1024$. *Robust* models are obtained by grid search over $\eta$ and $R$ against $l_\infty(2/255)$, where Non-DP: $\eta = 0.0625$; $DP_{\epsilon=2}$: $\eta = 4, R = 0.0625$; $DP_{\epsilon=4}$: $\eta = 0.5, R = 0.0625$, $DP_{\epsilon=8}$: $\eta = 0.125, R = 0.25$. Similarly, adversarial training models are obtained by grid search over $\eta$ and a fixed $R = 0.0625$, where Non-DP: $\eta = 0.25$; $DP_{\epsilon=2}$: $\eta = 0.5$; $DP_{\epsilon=4}$: $\eta = 1$, $DP_{\epsilon=8}$: $\eta = 2$. Other settings are the same as the *natural* ones. Adversarial attack is $l_\infty$, 20 steps, alpha $= 0.1$.

In Table 3, SimCLRv2 models are pre-trained on unlabelled ImageNet and fine-tuned on CIFAR10. *Natural* models are directly adopted from Tramer & Boneh (2020) for highest natural accuracy, where optimizer is DP-SGD and SGD, $R = 0.1$, $\eta_{DP} = 4$, $\eta_{non-DP} = 0.4$, momentum $= 0.9$, batch size $= 1024$. *Robust* models are obtained by grid search over $\eta$ and $R$ against $l_2(0.25)$, where Non-DP: $\eta = 0.0625$; $DP_{\epsilon=2}$: $\eta = 0.0625, R = 0.25$; $DP_{\epsilon=4}$: $\eta = 0.5, R = 0.0625$, $DP_{\epsilon=8}$: $\eta = 0.125, R = 0.25$. Similarly, adversarial training models are obtained by grid search over $\eta$ and a fixed $R = 0.0625$, where Non-DP: $\eta = 0.0625$; $DP_{\epsilon=2}$: $\eta = 1$; $DP_{\epsilon=4}$: $\eta = 1$, $DP_{\epsilon=8}$: $\eta = 2$. Other settings are the same as the *natural* ones. Adversarial attack is $l_2$, 20 steps, alpha $= 0.1$.

In Figure 6, models are SimCLRv2 pre-trained on unlabelled ImageNet and fine-tuned on CIFAR10 using DP-SGD, with $\epsilon = 8$, batch size $= 1024$. Adversarial attack is $l_\infty$ PGD, $\gamma = 4/255$, alpha=0.1.

In Table 4, models the same as in Table 2 with *robust* hyper-parameters, where optimizer is DP-SGD and SGD, momentum $= 0.9$, batch size $= 1024$, Non-DP: $\eta = 0.0625$; $DP_{\epsilon=2}$: $\eta = 4, R = 0.0625$; $DP_{\epsilon=4}$: $\eta = 0.5, R = 0.0625$, $DP_{\epsilon=8}$: $\eta = 0.125, R = 0.25$. Adversarial attack steps $= 20$, alpha $= 0.1$ if applicable.

In Table 5, models are ResNet18 and ViT-tiny trained on CelebA, label Smiling. Images are resized to $224 \times 224$. Optimizer is DP-RMSprop with epochs $= 5$, batch size $= 1024$, $\eta = 0.0002$, $R = 0.1$, delta=5e-6. Adversarial attack is $l_\infty$ PGD, 20 steps, alpha $= 1/255$.

In Table 6, models are ResNet18 as in Table 5, trained on CelebA, label 'Smiling'. Images are resized to $224 \times 224$. Optimizer is DP-RMSprop with epochs $= 5$, batch size $= 1024$, $\eta = 0.0002$, $R = 0.1$, delta=5e-6. Adversarial attack is $l_\infty(2/255)$ with $alpha_\infty = 1/255$ and $l_2(0.25)$ with $alpha_2 = 0.2$, 20 steps, if applicable.

In Figure 13, models are 2-layer CNN trained on CelebA label 'Male' using DP-Adam, where $\epsilon = 2$, batch size $= 512$, epochs $= 10$. Adversarial attack is $l_\infty(2/255)$ PGD, 20 steps, alpha $= 0.1$.

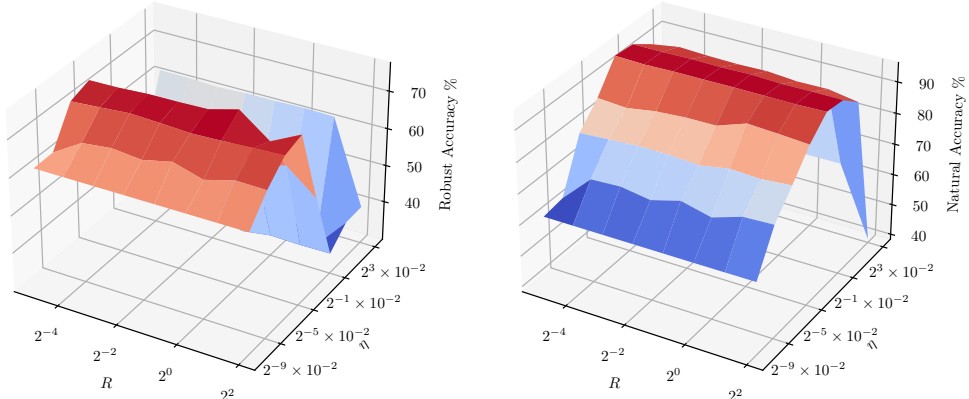

Figure 13: Robust and natural accuracy by $\eta$ and $R$ on CelebA with label 'Male'. We train a simple CNN with DP-Adam and test under 20 steps of $l_\infty(2/255)$ PGD attack. See details in Appendix D.

# E  Extra

## E.1  Robust and accuracy landscapes of DP optimizers

We note that the diagonal pattern of the accuracy landscapes observed in Figure 5 (CIFAR10 & DP-Heavyball), Figure 10 (CIFAR10 & DP-SGD), Figure 11 (MNIST & DP-SGD) and Figure 12 (Fashion MNIST & DP-SGD) is not universal. For example, in Figure 7, we show that adaptive optimizers are much less sensitive to the clipping norm $R$, as the landscapes are characterized by the row-wise pattern instead of the diagonal pattern. This pattern is particularly obvious in the small $R$ regime, where the robust and natural accuracy are high (see right panel in Figure 7.).

To rigorously analyze the insensitivity to the clipping norm in a simplified manner, we take the RMSprop(Tieleman et al., 2012) as an example, similar to the analysis in (Bu et al., 2023b) on DP-Adam. When $R$ is sufficiently small, the private gradient in (3) becomes

$$\tilde{\boldsymbol{g}}_t = \sum_i \frac{\boldsymbol{g}_t(\boldsymbol{x}_i)}{\max(1, ||\boldsymbol{g}_t(\boldsymbol{x}_i)||_2/R)} + \sigma R \mathcal{N}(0, I) = \sum_i \frac{\boldsymbol{g}_t(\boldsymbol{x}_i)}{||\boldsymbol{g}_t(\boldsymbol{x}_i)||_2/R} + \sigma R \mathcal{N}(0, I)$$

$$= R \cdot \left( \sum_i \frac{\boldsymbol{g}_t(\boldsymbol{x}_i)}{||\boldsymbol{g}_t(\boldsymbol{x}_i)||_2} + \sigma \mathcal{N}(0, I) \right) := R \cdot \hat{\boldsymbol{g}}_t.$$

With private gradient $\tilde{\boldsymbol{g}}_t$, DP-RMSprop updates the parameters $\boldsymbol{\theta}_t$ by

$$\boldsymbol{\theta}_t = \boldsymbol{\theta}_{t-1} + \eta_t \frac{\tilde{\boldsymbol{g}}_t}{\sqrt{\tilde{\boldsymbol{v}}_t}}, \tag{9}$$

where $\tilde{\boldsymbol{v}}$ is the squared average of $\tilde{\boldsymbol{g}}_t$, written as

$$\tilde{\boldsymbol{v}}_t = \alpha \tilde{\boldsymbol{v}}_{t-1} + (1-\alpha)\tilde{\boldsymbol{g}}_t^2 = \sum_s^t (1-\alpha)\alpha^{t-s}\tilde{\boldsymbol{g}}_s^2 = R^2 \cdot \sum_s^t (1-\alpha)\alpha^{t-s}\hat{\boldsymbol{g}}_s^2 \tag{10}$$

Substitute (10) into (9), we obtain an updating rule that is independent of the clipping norm $R$,

$$\boldsymbol{\theta}_t = \boldsymbol{\theta}_{t-1} + \eta_t \frac{R \cdot \hat{\boldsymbol{g}}_t}{\sqrt{R^2 \cdot \sum_s^t (1-\alpha)\alpha^{t-s}\hat{\boldsymbol{g}}_s^2}} = \boldsymbol{\theta}_{t-1} + \eta_t \frac{\hat{\boldsymbol{g}}_t}{\sqrt{\sum_s^t (1-\alpha)\alpha^{t-s}\hat{\boldsymbol{g}}_s^2}}.$$

As a result, DP optimizers can have fundamentally different landscapes with respect to the hyper-parameters $(R, \eta)$, which in turn may affect the accuracy and robustness as illustrated in Figure 8.

