# OpenReview forum: "Differentially Private Optimizers Can Learn Adversarially Robust Models"
_TMLR — Accepted by TMLR_

### Review · Reviewer_MWSt · 2023-09-01

**Summary Of Contributions:**

The paper provides an analytic and empirical study to understand whether the DP-trained models could be adversarially robust. Theoretically, the authors study the optimal adversarially robust errors that can be achieved by private linear classifiers (where the DP noise is only applied to the bias term fine-tuning) under certain conditions. Empirically, authors find out that with proper tuning of key hyperparameters (e.g., learning rate and clipping norm), DP-trained deep neural networks could also be more adversarially robust than the non-private ones.

**Audience:**

Yes

**Claims And Evidence:**

Yes

**Requested Changes:**

1. The deviation of the last line in Eq. (8) is still unclear to me, e.g., why there is K in the second term? Could you please explain the deviation in more detail (e.g., how to derive the function input of CDF of the standard normal)?

2. Why do you choose (R, $\eta$) as the choice of hyperparameters for your experiment? How do other hyperparamters (e.g., batch size) affect the performance? Please justify this point in the paper.

3. Is this a typo “76.13% accuracy can degrade to 0.00% accuracy?”

**Strengths And Weaknesses:**

Strengths
1.  This paper answers a very important question: Can a model be simultaneously private and robust? Besides, the authors also provide concrete empirical suggestions on how to train a robust DP model (e.g., using small clipping norms and large learning rates). The results are very important since they could guide the future training of DP robust models.
2. The paper is technically sound, and the experiments are extensive.

Weaknesses
1. Technical novelty is limited. Some proofs (e.g., Theorem 1) are direct extensions from Xu et al. (2021).
2. There is a disconnect between the theory and experiments. For example, the DP mechanism used in the theoretical analysis is bias term fine-tuning, while the experiments use DP-SGD or DP-Adam.
3. Presentation/Clarity needs improvement. (See requested changes below)

---

> ### Author Response · Authors · 2023-09-21
>
> We thank the reviewer for the comments!
>
> **Technical novelty is limited. Some proofs (e.g., Theorem 1) are direct extensions from Xu et al. (2021).**
>
> Response: We agree that some proofs are extended from the enlightening work of Xu et al. (2021). However, the technical novelty of our work is non-trivial. We would like to point out that our Theorem 2 (DP model can be the most robust, with a specific analysis on how per-sample gradient clipping helps in Figure 4) and Theorem 3 (DP models are Pareto optimal compared to non-DP models) are new and calibrated to DP.
>
> **There is a disconnect between the theory and experiments.**
>
> Response: We agree that the experiments are much richer than the theory, in order to cover the more interesting and practical settings in deep learning. In this work, we chose to do analysis on simple models in order to gain deep insights, rather than on complex models which are vastly more difficult. Typically, to analyze deep learning models with DP, we have to rely on hard-to-justify assumptions and compromise to weaker conclusions. We hope the reviewer would agree that a disconnect is the norm in deep learning paper (including Xu et al. 2021), which can serve as the motivation for future work.
>
> **The deviation of the last line in Eq. (8) is still unclear to me, e.g., why there is K in the second term? Could you please explain the deviation in more detail (e.g., how to derive the function input of CDF of the standard normal)?**
>
> Response: Sure! K is introduced in the setting in Eq. (5), which can be viewed as the difficulty to learn the positive class samples (c.f. Footnote 2). Therefore, the second term in the second last line of Eq. (8) involves K:
>
> $P(\sum_j w_j(x_j-\gamma)+b<0|y=+1)=P(N(d(\theta-\gamma)+b,dK^2\sigma^2)<0)=\Phi(-\frac{d(\theta-\gamma)+b}{\sqrt{d}K\sigma})$
>
> where $x_j-\gamma\sim N(\theta-\gamma,K^2\sigma^2)$ is i.i.d. for each $j$ from Eq.(5), and thus we used $\sum_j x_j-\gamma\sim N(d(\theta-\gamma),dK^2\sigma^2)$. We will surely add this in appendix.
>
> **Why do you choose (R, $\eta$) as the choice of hyperparameters for your experiment? How do other hyperparamters (e.g., batch size) affect the performance? Please justify this point in the paper.**
>
> Response: We choose R and $\eta$ because they are only related to optimization but not to privacy. This property allows us to decouple their effects on robustness. Put differently, if we change batch size, then either the number of training iterations or noise level $\sigma_{DP}$ need to change accordingly to satisfy the same $(\epsilon,\delta)$ DP guarantee. We discussed this in the first paragraph of Sec 4.1. In summary, we show that, without affecting DP mechanism, changing hyperparameters can result in robust DP models.
>
> **Is this a typo “76.13% accuracy can degrade to 0.00% accuracy?”**
>
> Response: This is not typo. Deep learning models are indeed very vulnerable to adversarial attacks, resulting in almost 0% accuracy. We attach a few links for your reference:
>
> 1. https://github.com/Harry24k/adversarial-attacks-pytorch#performance-comparison
>
> 2. https://github.com/MadryLab/robustness#pretrained-models

---

### Review · Reviewer_sCkX · 2023-09-04

**Summary Of Contributions:**

This paper provides the benefit of a differential private optimizer in the model robustness. Specifically, previous research showed that differentially private models might become more vulnerable to adversarial attacks. The authors challenge this previous finding and show that differentially private classifiers can achieve optimal robustness under some conditions.

The authors first provide a formal analysis on a linear classifier. The authors first compute the intercept for the optimal robust linear classifier in a mixed Gaussian distribution setting. Then, they show that differentially private optimization can achieve this intercept under some conditions. The authors also support the validity of the conditions with several experiments. Finally, the authors show that the differentially private training has better robustness for fixed attack radius than the natural training under some conditions.

Then, with experiments, the authors extend their findings on a linear classifier to more general cases. The authors first explore two parameters, i.e., clipping radius and learning rate, and check that differentially private training may show better robustness with a proper parameter setting. Then, the authors demonstrate the better robustness of differentially private models than their naturally trained counterparts. Next, the authors explore the performance of differentially private models against different attack methods. Finally, the authors present the experiments on a larger dataset.

**Audience:**

Yes

**Broader Impact Concerns:**

I don’t see a particular broader impact concern regarding this paper.

**Claims And Evidence:**

Yes

**Requested Changes:**

1. I suggest making the paper easier to read. The following would improve the paper presentation.
  - Write a “Related Works” section. The Related Works section helps the readers to understand the motivation by reading an overview of the research works related to the problem.
  - Put a summary of experiment goals at the beginning of Section 4. A short list of research questions that the experiments demonstrate will help the readers understand the purpose of the experiments.
  - The authors should put more details about the experiment setup in Section 4, rather than putting details in the captions of figures.
  - Reduce the redundancies in writing.
  - Have a proper “Conclusion” section. The first paragraph in the Discussion section does not sufficiently summarize the authors’ efforts.
2. Improve the experiments to strengthen the paper.
  - As mentioned in the weakness, the authors can use more attacks to support the insights better. Try attacks with different optimizations, e.g., CW attack, DeepFool, or gradient-free attacks, e.g., HopSkipJump.
  - Add comparisons to adversarial training methods and analyze the robustness gap (between a differentially private model and an optimal robust model) in practice.
3. Minor revisions
  - There is nearly no margin between Table 1 and the text part. Please fix.
  - In Figure 4, change “balanced” to ”left” and “unbalanced” to ”right”, or just put titles to each subplot. Also, the x_label in the left subplot seems incorrect; the gradient is missing inside the clipping.
  - Does Corollary 3.2 require the clipping to change from $l_\infty$ to $l_2$? If so, include details about the necessary changes and clarify that we need different DP optimization for different clipping norms. If not, write a discussion about why such modification is unnecessary.
  - Theorems are numbered with 1,2,3…, but all the others are numbered with section numbers. Change Definition 2.1 to Definition 1, Fact 3.1 to Fact 1, and Corollary 3.2 to Corollary 1.
  - To the best of my knowledge, “et al.” is not supposed to be used when all the author names are listed. Check the bib file and remove the uses of “et al.” in the bibliography.

**Strengths And Weaknesses:**

# Strengths
1. The authors performed the experiments in many different settings: used three other datasets (including the experiments in the Appendix), two metric norms, three target models, and at least four different attacks.
2. The theoretical analysis seems valid and meaningful.

# Weaknesses
1. The authors should test more attacks in the experiments. While the authors used four different attacks, i.e., FGSM, BIM, PGD, and APGD, in the experiments, all those attacks work similarly. To explain, all those methods compute gradients of the objective function, then compute the sign of the gradient to find the next direction in each step: They are different only in the number of steps, or random, starts or clipping, or step sizes. They use fundamentally similar optimization methods with slight differences in the details of each step.
2. The authors only compared differentially private training to natural training. However, it is better to contain a comparison to adversarial training methods. While I don’t expect the differentially private training to have better robustness than the state-of-the-art adversarial training methods, it is still worth presenting the robustness gap between a differentially private model and optimal robust models. (Of course, the adversarial training is not optimal, but the comparison will show the lower bound of the gap.)

---

> ### Author Response · Authors · 2023-09-23
>
> We thank the reviewer for the helpful comments! Please see the revision for our changes.
>
> We have fixed the margin between Table 1 and the text, changed Figure 4 caption and x-label, and re-numbered definitions, facts and corollaries. We note that the bibliography is automatic and following TMLR style. We have made sure that there is no “et al.” in the bibliography but happy to make further changes if you can kindly point out some specifics.
>
> **Put a summary of experiment goals at the beginning of Section 4. A short list of research questions that the experiments demonstrate will help the readers understand the purpose of the experiments.**
>
> We have added the summary of experiment goals as suggested. Thank you for this suggestion!
>
> **The first paragraph in the Discussion section does not sufficiently summarize the authors’ efforts.**
>
> We have added more discussion in the last section to summarize our efforts and conclusions.
>
> **The authors should test more attacks in the experiments. While the authors used four different attacks, i.e., FGSM, BIM, PGD, and APGD, in the experiments, all those attacks work similarly. To explain, all those methods compute gradients of the objective function, then compute the sign of the gradient to find the next direction in each step..... Try attacks with different optimizations, e.g., CW attack, DeepFool, or gradient-free attacks, e.g., HopSkipJump.**
>
> We would like to point out that the attacks evaluated in this work are not fundamentally similar. Take FGSM and PGD as an example: FGSM computes the sign of gradient (so each element in gradient is +1 or -1) and only attacks the input for one time in one iteration (that is, each time the model is updated, the adversarial input is updated once); PGD **does not compute the sign of gradient** because it projects the gradient to l2 or l_infty ball (so each element in gradient is a floating number). PGD attacks the input multiple times in one iteration (that is, each time the model is updated, the adversarial input has been updated 50 times).
>
> We have selected commonly used attacks that cover single-step (FGSM) v.s. multiple-step (PGD), $l_\infty$ (PGD$_\infty$) v.s. $l_2$ (PGD$_2$), and single-method (PGD) v.s. ensemble (APGD). Therefore, although including more attacks would strengthen the paper, we believe that our current selection is sufficient. Particularly, one major contribution is our theoretical analysis, which is attack-independent.
>
> **Add comparisons to adversarial training methods and analyze the robustness gap (between a differentially private model and an optimal robust model) in practice.**
>
> We note that Theorem 2 (if extended to deep learning) states that DP non-adversarially-trained model can be the optimal robust for some attack level. However, this level may be too small to be useful, e.g. only allowing one to manipulate a sample by 1/10000. We believe Theorem 3 is more insightful and practical: we may expect DP non-adversarially-trained model $\geq$ natural (non-adversarially-trained) model in terms of robustness, as we have extensively reported in Table 1-6.
>
> As a consequence, a fair comparison would be DP adversarially-trained model v.s. non-DP adversarially-trained model. This has been evaluated in some works and hence out of scope of this paper.
>
> **Does Corollary 3.2 require the clipping to change from $l_\infty$ to $l_2$? If so, include details about the necessary changes and clarify that we need different DP optimization for different clipping norms. If not, write a discussion about why such modification is unnecessary.**
>
> We emphasize that in DP deep learning (e.g. Abadi et al. 2015) the per-sample clipping is always l2 **on the gradient level** whereas Corollary 3.2 (now Corollary 1 after re-numbering) refers to the attack **on the sample/data level**. Our l2 clipping was clearly stated in Footnote 1. We follow your suggestion and add a discussion in Footnote 5.

---

### Review · Reviewer_xGht · 2023-09-14

**Summary Of Contributions:**

This paper argues that a differentially private machine learning model is actually a adversarially robust as well. The author prove this for a special case where the data comes from gaussian mixture model and the model has only one parameter. The authors also show that their theoretical results can be applied to fine-tuning of deep models through empirical study.

**Audience:**

Yes

**Broader Impact Concerns:**

No Concern related  to Broader Impact Concerns

**Claims And Evidence:**

Yes

**Requested Changes:**

1. Is it possible to provide theoretical analysis for a scenario that $w$ is not fixed before hand and needs to be trained/learned?

2. Could you please clarify why theorem 2 implies robustness. I feel $\gamma*$ can be arbitrary small.

3. Please clarify how the privacy parameters $(\epsilon,\delta)$ have been calculated? The reported privacy loss is for the whole training process or for only one iterations?

**Strengths And Weaknesses:**

Strengths:
1. The authors start with a very simple linear model and they provide theoretical study on that. This simple model makes it easy for the reader to understand the main message of the paper.

2. The authors go beyond the linear model and they show their claim through extensive empirical study on non-linear models.

Weaknesses:

1. There is no theoretical analysis beyond a model with only one parameter (parameter $b$)
2. Theorem 2 somehow is misleading. It shows that there is a $\gamma*$ for each $b_{DP}$ such that the model is robust to  attack level $\gamma*$. And then the authors claim that this theorem implies robustness. However, there no lowerbound on $\gamma*$. It can be very small. If $\gamma*$ is close to zero, technically we have a very weak robustness.
3. In the numerical experiment part, it is not clear how the privacy loss has been calculated. For DP-SGD, the parameter $\epsilon$ depends on batch size, number of iterations. Technically, there is no closed form for $\epsilon$ as a function of batch size, number of iterations. The privacy curve has to be estimated numerically. The authors also do not report $\delta$ in the tables.

---

> ### Author Response · Authors · 2023-09-21
>
> We thank the reviewer for the comments!
>
> **Weakness 1: There is no theoretical analysis beyond a model with only one parameter.**
>
> Response: We agree it would be desirable to analyze complex models. In this work, we chose to do analysis on simple models in order to gain deep insights, rather than on complex models which are vastly more difficult. Typically, to analyze deep learning models with DP, we have to rely on hard-to-justify assumptions and compromise to weaker conclusions.
>
> **Weakness 2: Theorem 2 somehow is misleading...However, there no lowerbound on $\gamma^\*$. It can be very small. If is close to zero, technically we have a very weak robustness.**
>
> Response: You are right that if $\gamma^\*$ is close to zero, then the robustness is weak. However, this is not a weakness of DP because all linear models $f$ at most guarantee the weak robustness (see the $min_f$ in Theorem 2). That is, for this $\gamma^\*$, DP linear model is relatively the strongest, though it is still weak. We would make sure this discussion is added to the main text.
>
> **Weakness 3: In the numerical experiment part, it is not clear how the privacy loss has been calculated...The authors also do not report
>  in the tables.**
>
> Response: We thank the reviewer for pointing this out. Let us firstly clarify that the delta is stated in Appendix D. Now for the epsilon, we were using Renyi accountant as is the standard tool in [1][2][3] and offered in popular codebases such as Opacus and FastDP. We will add the description in the main text. On a side note, the choice of privacy accountant may have small differences in the robustness and accuracy, e.g. if we instead use PRV accoutant in "Numerical Composition of Differential Privacy", but will not change the pattern that DP models can be robust.
>
> **Is it possible to provide theoretical analysis for a scenario that w is not fixed before hand and needs to be trained/learned?**
>
> Response: We believe it is possible but additional assumptions may be needed to quantify the distance between $w_{DP}$ and $w_{non-DP}$.
>
> **Could you please clarify why theorem 2 implies robustness. I feel $\gamma^\*$ can be arbitrary small.**
>
> Response: Theorem 2 says that for a fixed DP model, one may find a $\gamma^\*$ robustness level (which could be small) such that this DP model is the strongest among all linear models. The requirement that "DP model is the strongest" may be too strict so we have Theorem 3 that says for a fixed robustness level $\gamma$ (which could be big), one may find a DP model that is not the strongest but at least stronger than the non-DP model that is the most naturally accurate.
>
> **Please clarify how the privacy parameters have been calculated? The reported privacy loss is for the whole training process or for only one iterations?**
>
> Response: The privacy parameters are calculated using Renyi accountant (as in the Opacus codebase) for the whole training process.
>
> [1]  Large language models can be strong differentially private learners
>
> [2] Automatic clipping: Differentially private deep learning made easier and stronger.
>
> [3] Unlocking High-Accuracy Differentially Private Image Classification through Scale

---

### Decision · Action_Editor_6fBG · 2023-10-24

**Recommendation:** Accept with minor revision

**Comment:**

This paper has mixed opinions from the reviewers, while the positive opinions slightly outweigh the negative opinions. I carefully read the entire reviews, corresponding responses, and the revised paper. As my comment in "Claims And Evidence", both the theoretical and empirical results have some flaws. I think this paper needs a minor revision to support its claim better. Although I requested additional contributions, I presume that they will not weaken the contribution of the paper because (1) the authors claimed that the extension would be possible by introducing additional assumptions (2) even though AT models perform better in terms of accuracy-robustness trade-off the main theorem does not argue about the relationship between AT models and DP models. However, if AT models perform better than DP models, I think the main argument should be fixed to "DP models are theoretically and empirically better than naturally-trained (non-DP) models in terms of the accuracy-robustness tradeoff".

**Audience:**

Both differential privacy (DP) and adversarial robustness are the central topics of trustworthy AI. As the paper pointed out, previous studies, such as Tursynbek et al. (2020) and Boenisch et al. (2021), showed that DP models show more adversarial vulnerability than naturally-trained models, while leaving a question "DP leads to adversarial vulnerability"? This paper theoretically and empirically shows that it may not be true. Although the theoretical results are limited and empirical studies need more improvements, I think we can easily find audiences interested in this paper.

**Claims And Evidence:**

The main claim of this paper is "DP models are theoretical and empirical Pareto optimal on the accuracy-robustness tradeoff". This paper provides both theoretical and empirical results to support the claim.

The theory part, as all the reviewers pointed out, is somewhat weak. First, theoretical novelty can be somewhat weakened as the idea highly relies on Xu et al. 2021. It does not mean that theoretical contribution is "trivial". As the authors claimed, although the part of the theorem is an extension of Xu et al. 2021, this submission has its own contribution to bring the theorem to a DP scenario. Second, the gap between the scenario introduced for the theory and the actual learning scenario is too large; the theory assumes a linear classifier where $w$ is fixed and only $b$ is trained, but the training scenario is based on complex deep neural networks. I partially agree with the claim by the authors: "Typically, to analyze deep learning models with DP, we have to rely on hard-to-justify assumptions and compromise to weaker conclusions.". However, as the authors mentioned, if it is possible to extend the theorem beyond the fixed $w$ (with more assumptions), I think it would be better to extend the theorem. Finally, as Reviewer xGht mentioned, the theorem could lead to a trivial bound because there is no assumption of $\gamma$. The authors rebutted this by arguing that DP models are still "relatively stronger" than natural models; I think it would be better to provide more discussion of $\gamma$, but I also understand that it could be very difficult.

In the empirical analysis part, I agree with the initial review by Reviewer sCkX: "it is better to contain a comparison to adversarial training methods". It is because the main argument of this paper is "DP models are theoretical and empirical Pareto optimal on the accuracy-robustness tradeoff". This paper only compares DP models and natural models and keeps arguing that a DP model is better than a naturally trained model. As there was no guarantee of the relationship between DP and adversarial robustness, I think adversarial training and DP also have no theoretical guarantee. As this paper argues "Pareto optimal", I think adversarially-trained models should be compared to the baselines.

Overall, this paper supports the main claim theoretically and empirically, while both have some flaws. I think this paper is intriguing enough to TMLR audiences, but it would be better to improve the submission with the following options.

1. Extending the theorem for the non-fixed $w$ case as Reviewer xGht's comment: This comment is based on the authors' response: "We believe it is possible but additional assumptions may be needed". It will need a revision of the manuscript. If it is non-trivial to show the non-fixed $w$ scenario, then the paper should discuss in details why it is non-trivial.
2. Add comparisons with non-DP adversarially trained (AT) models. It is because the main claim of this paper is "DP models are theoretical and empirical Pareto optimal on the accuracy-robustness tradeoff". I think it would be enough with the simplest version of AT models, such as PGD-trained AT models without pre-training. Note that AT models usually use very wide networks, such as WideResNet, I think it is enough for showing AT models trained on the same architecture by the other baselines. If the authors think this is still out-of-scope, I strongly suggest to tone down the argument "DP models are theoretical and empirical Pareto optimal on the accuracy-robustness tradeoff" to "DP models are theoretical and empirical better than naturally-trained (non-DP) models in terms of the accuracy-robustness tradeoff". Any of them will need a revision of the manuscript.

---

> ### Author Response · Authors · 2023-11-16
> **Revision submitted.**
>
> Dear Editor,
>
> Thank you for handling our paper! We have carefully revised our paper to accommodate the comments from you and the reviewers. A detailed list of changes can be found in "Changes Since Last Submission". Here is a brief summary:
>
> We have added a new paragraph *explaining the difficulty of non-fixed $w$ scenario from an optimization viewpoint*. The problem imposed by DP will be constrained and hard to solve. We have added new experiments in Table 2 & 3 to *compare with adversarial training (with or without DP)*, which empirically supports our theorems by extending to the non-fixed $w$ scenario and to the non-convex deep learning regime. We have consistently observed that DP models can be adversarially robust.
>
> Please kindly let us know if further revision is needed for the camera-ready version as it is.